# Comparing Few to Rank Many: Active Human Preference Learning Using Randomized Frank-Wolfe Method

Kiran Thekumparampil [1]   Gaurush Hiranandani [2]   Kousha Kalantari [1]   Shoham Sabach [1 3]   Branislav Kveton [4]

## Abstract

We study learning human preferences from limited comparison feedback, a core machine learning problem that is at the center of reinforcement learning from human feedback (RLHF). We formulate the problem as learning a Plackett-Luce (PL) model from a limited number of $K$-subset comparisons over a universe of $N$ items, where typically $K \ll N$. Our objective is to select the $K$-subsets such that all items can be ranked with minimal mistakes within the budget. We solve the problem using the D-optimal design, which minimizes the worst-case ranking loss under the estimated PL model. All known algorithms for this problem are computationally infeasible in our setting because we consider exponentially many subsets in $K$. To address this challenge, we propose a randomized Frank-Wolfe algorithm with memoization and sparse updates that has a low $O(N^2 + K^2)$ per-iteration complexity. We analyze it and demonstrate its empirical superiority on synthetic and open-source NLP datasets.

## 1. Introduction

Learning to rank from human feedback is a fundamental machine learning problem. We consider the setting where a larger set of $N$ items is ranked based on feedback on its $K$-subsets. This setting has various practical applications. In web search (Agichtein et al., 2006; Hofmann et al., 2013; 2016), the items are retrieved web pages for a given query and they are rated by human annotators. Since this task is time consuming, the annotators typically label only a subset of the web pages (Hofmann et al., 2013; MS MARCO). Online marketplaces display only a handful of items from a catalogue of billions of items to a user and the user's choice among them can be viewed as a noisy ranking observation.

These real-world interactions are well-modeled by preference models studied in this work (Negahban et al., 2018). Finally, in *reinforcement learning from human feedback (RLHF)* (Ouyang et al., 2022; Rafailov et al., 2023), human feedback is used to learn a reward model, which is then used to align a *large-language model (LLM)* with human preferences. In this case, the items are LLM responses and the relative feedback is used because it yields better alignment than absolute feedback (Christiano et al., 2017; Bai et al., 2022). A comparison of more than two responses has been investigated because it is practical and improves statistical efficiency (Zhu et al., 2023b; Mukherjee et al., 2024).

We formulate the problem as learning to rank $N$ items, such as web pages or LLM responses, from a limited $K$-way feedback. In general, $K \ll N$ because humans cannot provide high-quality preferential feedback on a large number of choices (Tversky & Kahneman, 1974). When $K = 2$, we get a relative feedback over two responses, known as the *Bradley-Terry-Luce (BTL)* model (Bradley & Terry, 1952). When $K \geq 2$, we get a ranking feedback over $K$ responses, known as the *Plackett-Luce (PL)* model (Plackett, 1975; Luce, 2005). To learn high-quality preference models, we ask humans questions that maximize information gain. Such problems have long been studied in the field of optimal design (Kiefer & Wolfowitz, 1960; Pukelsheim, 2006). An optimal design is a feedback collection policy that minimizes the uncertainty in the downstream learning task. This policy can be learned by iterative algorithms, such as the *Frank-Wolfe (FW)* method (Khachiyan, 1996). The main challenge of our setting is that these algorithms are impractical when $K \ll N$, since we need to optimize over exponentially-many $O(\binom{N}{K})$ $K$-subsets. In this work, we focus on solving large-scale optimal designs with trillions of potential choices for $K$-way feedback. We make the following contributions:

**(1)** We propose a general optimal design framework for collecting human feedback to learn to rank $N$ items from $K$-way feedback (Sections 2 and 3), where $K \leq N$. This generalizes known frameworks (Mukherjee et al., 2024; Mehta et al., 2023; Das et al., 2024) that assume $K = N$.

**(2)** We bound the prediction error and ranking loss of the PL model learned from human feedback (Section 3.3). Both

---

[1]Amazon [2]Typeface [3]Technion [4]Adobe Research. Correspondence to: Kiran Thekumparampil <kkt@amazon.com>.

decrease as the amount of feedback increases.

**(3)** We propose a novel algorithm `DopeWolfe` for solving our optimal design efficiently (Section 5). `DopeWolfe` has a low $O(N^2 + K^2)$ per-iteration complexity. In contrast, the per-iteration complexity of the FW method is $O(\binom{N}{K})$. We achieve this with the help of a randomized FW method (Zhao & Freund, 2023), combined with memoization and sparse operations.

**(4)** We provide a convergence analysis for `DopeWolfe` by generalizing the analysis of the randomized FW method beyond Lipschitz smoothness (Kerdreux et al., 2018) to a general class of logarithmically homogeneous self-concordant barrier (LHSCB) objectives (Theorem 3 and Corollary 4).

**(5)** We empirically evaluate `DopeWolfe` on both synthetic and real-world problems (Section 6). We observe an order of magnitude reduction in sample size in our experiments on the Nectar dataset (Zhu et al., 2023a).

Our real-world datasets cover reward modeling in RLHF and learning to rank. However, we stress that all our experiments are on linear models over frozen language model embeddings instead of fine-tuning LLMs. In this way, we stay faithful to the Plackett-Luce model that we study.

**Related work.** There are two lines of related works: learning to rank from human feedback and solving large-scale optimal designs. Mehta et al. (2023) and Das et al. (2024) learn to rank pairs of items from pairwise feedback. They optimize the maximum gap. Mukherjee et al. (2024) learn to rank lists of $K$ items from $K$-way feedback. They optimize both the maximum prediction error and ranking loss. We propose a general framework for learning to rank $N$ items from $K$-way feedback, where $K \geq 2$ and $N \geq K$. This setting is more general than in the prior works. Our main algorithmic contribution is a randomized Frank-Wolfe algorithm. It addresses a specific computational problem of our setting, solving a D-optimal design with $O(\binom{N}{K})$ variables for learning to rank. This distinguishes it from other recent works for solving large-scale optimal designs (Hendrych et al., 2023; Ahipaşaoğlu, 2015; Zhao & Freund, 2023). We review related works in more detail in Appendix A.

This paper is organized as follows. Section 2 introduces our ranking problem and Section 3 presents our framework for collecting human feedback to rank $N$ items using $K$-way feedback. We show the impracticality of the FW method in Section 4 and make it efficient in Section 5. Section 6 is devoted to experiments. We conclude in Section 7.

## 2. Setting

We start with introducing notation. Let $[n] = \{1, \dots, n\}$ and $\mathbb{R}^d$ be a $d$-dimensional real vector space. Let $\triangle^S$ be the probability simplex over set $S$. For any distribution

$\pi \in \triangle^S$, $\sum_{i \in S} \pi(i) = 1$. For any positive-definite matrix $A \in \mathbb{R}^{d \times d}$ and vector $x \in \mathbb{R}^d$, let $\|x\|_A = \sqrt{x^\top A x}$. We say that a square matrix $M \in \mathbb{R}^{d \times d}$ is PSD / PD if it is positive semi-definite / positive definite.

We study the problem of learning to rank $N$ items. An item $k \in [N]$ is represented by its feature vector $x_k \in \mathcal{X}$, where $\mathcal{X} \subseteq \mathbb{R}^d$ is the support of all feature vectors. The relevance of item $k$ is defined as $x_k^\top \theta_*$, where $\theta_* \in \mathbb{R}^d$ is an unknown parameter, and we call it the *mean reward*. Linear models of relevance have a long history in learning to rank (Zong et al., 2016; Li et al., 2016). A standard way of applying them to modern machine learning problems is to set $x_k$ to the last hidden-layer embedding of the neural network (Riquelme et al., 2018). Our experiments in Section 6 are with such embeddings. Without loss of generality, we assume that the original order of the items is optimal, $x_j^\top \theta_* > x_k^\top \theta_*$ for any $j < k$.

We interact with a human $T$ times. In interaction $\ell \in [T]$, we select a $K$-subset of items $S_\ell \in \mathcal{S}$ and the human ranks all items in $S_\ell$ according to their preferences, where $\mathcal{S}$ is a collection of all $K$-subsets of $[N]$. Note that $|\mathcal{S}| = \binom{N}{K}$. We represent the ranking as a permutation $\sigma_\ell : [K] \to S_\ell$, where $\sigma_\ell(k)$ is the item at position $k$. The probability that this permutation is generated is

$$p(\sigma_\ell) = \prod_{k=1}^K \frac{\exp[x_{\sigma_\ell(k)}^\top \theta_*]}{\sum_{j=k}^K \exp[x_{\sigma_\ell(j)}^\top \theta_*]} . \qquad (1)$$

In short, items with higher mean rewards are more preferred by humans and thus more likely to be ranked higher. This feedback model is known as Plackett-Luce (Plackett, 1975) and is the most popular approach to eliciting unknown human preferences from ranking multiple options (Negahban et al., 2018).

Our goal is to select the subsets such that we can learn the true order of the items. Specifically, after $T$ interactions, we output a permutation $\hat{\sigma} : [N] \to [N]$, where $\hat{\sigma}(k)$ is the index of the item at position $k$. The quality of the solution is measured by the *ranking loss*

$$R(T) = \frac{2}{N(N-1)} \sum_{j=1}^N \sum_{k=j+1}^N \mathbb{1}\{\hat{\sigma}(j) > \hat{\sigma}(k)\} , \quad (2)$$

where $N(N-1)/2$ is a normalizing factor that scales the loss to $[0, 1]$. Simply put, the ranking loss is the fraction of incorrectly ordered pairs of items in permutation $\hat{\sigma}$. It can also be viewed as the normalized Kendall tau rank distance (Kendall, 1948) between the optimal order of items and that according to $\hat{\sigma}$, multiplied by $2/(N(N-1))$. While other objectives are possible, such as the *mean reciprocal rank (MRR)* and *normalized discounted cumulative gain (NDCG)* (Manning et al., 2008), we focus on the ranking loss in (2)

to simplify our presentation. For completeness, we report the NDCG in our experiments.

To simplify exposition, and without loss of generality, we focus on ranking $N$ items in a single list. Our setting can be generalized to multiple lists as follows. Suppose that we want to rank multiple lists with $N_1, \ldots, N_M$ items using $K$-way feedback. This can be viewed as a ranking problem over $\left| \binom{N_1}{K} \right| + \cdots + \left| \binom{N_M}{K} \right|$ subsets, where any $K$-subset of items from the $M$ lists is included in $\mathcal{S}$. Since our algorithm depends only on $\mathcal{S}$ (Section 5), it remains the same even in this more challenging setting. In fact, we experiment with this setting in Section 6.2, where we learn to rank $30\,000$ lists, each with 7 items from $K$-subsets of size $K = 2$ and $K = 3$. Even for $K = 2$, this is $630\,000$ subsets and two orders of magnitude more than in prior works (Mukherjee et al., 2024).

# 3. Optimal Design for Learning To Rank

In this section, we introduce our basic algorithm and analyze it. More specifically, we generalize Mukherjee et al. (2024) to ranking $N$ items from $K$-way feedback, from ranking $K$ items only, for any $K \leq N$. While these contributions are not surprising technically, they significantly expand the class of modeled and solved problems. The fundamental novel challenge is solving the optimal design in (7). Since $|\mathcal{S}| = \binom{N}{K}$, it has exponentially many variables in $K$. Therefore, it cannot be written out and solved using standard methods, such as CVXPY (Diamond & Boyd, 2016). We address this challenge separately in Section 5.

## 3.1. Maximum Likelihood Estimate

We note that the problem of learning from $K$-way feedback is identical to that in Zhu et al. (2023b). Therefore, we just summarize their solution. Suppose that the human interacts with $T$ subsets $\{S_\ell\}_{\ell \in [T]}$ (Section 2). We use the human responses $\sigma_\ell$ on $S_\ell$ to estimate $\theta_*$. Specifically, since the probability of $\sigma_\ell$ under $\theta_*$ is (1), the negative log-likelihood of all feedback is

$$\mathcal{L}_T(\theta) = - \sum_{\ell=1}^{T} \sum_{k=1}^{K} \log \left( \frac{\exp[x_{\sigma_\ell(k)}^\top \theta]}{\sum_{j=k}^{K} \exp[x_{\sigma_\ell(j)}^\top \theta]} \right) . \quad (3)$$

To estimate $\theta_*$, we solve a *maximum likelihood estimation (MLE)* problem, $\hat{\theta} = \arg\min_{\theta \in \Theta} \mathcal{L}_T(\theta)$. The problem is convex (Zhu et al., 2023b) and thus can be solved efficiently by standard methods for *generalized linear models (GLMs)* (Bishop, 2006). Finally, we estimate the mean reward of item $k$ as $x_k^\top \hat{\theta}$ and sort the items in descending order of $x_k^\top \hat{\theta}$, which defines $\hat{\sigma}$ in (2).

## 3.2. Active Learning

The problem of choosing the most informative subsets $S_\ell$ for solving (3) was studied by Mukherjee et al. (2024) and can be summarized as follows. Let $\nabla^2 \mathcal{L}_T(\theta)$ be the Hessian of $\mathcal{L}_T(\theta)$, which can be used to measure the uncertainty in the MLE $\hat{\theta}$. Let $z_{j,k} = x_j - x_k$ be the difference of feature vectors of items $j$ and $k$, and $\mathcal{Z}$ denote the set of all feature vector differences. Let $z_{\ell,j,k} = z_{\sigma_\ell(j),\sigma_\ell(k)}$. In our notation, Zhu et al. (2023b) showed that a high-probability bound on the prediction error, for any $z \in \mathcal{Z}$, is

$$|z^\top(\hat{\theta} - \theta_*)| \leq \tilde{O}(\sqrt{d}\|z\|_{M^{-1}}), \quad (4)$$

where

$$M = \frac{C}{2K(K-1)} \sum_{\ell=1}^{T} \sum_{j=1}^{K} \sum_{k=j+1}^{K} z_{\ell,j,k} z_{\ell,j,k}^\top \quad (5)$$

is a lower bound on the Hessian, $\nabla^2 \mathcal{L}_T(\theta) \succeq M$ holds for any $\theta \in \Theta$ and a universal constant $C > 0$.

The minimization of $\max_{z \in \mathcal{Z}} \|z\|_{M^{-1}}$ is equivalent to maximizing $\log \det(M)$ (Kiefer & Wolfowitz, 1960). This maximization problem is known as the D-optimal design. Mukherjee et al. (2024) formulated it using matrix notation as follows. Each subset $S \in \mathcal{S}$ is represented by its matrix $A_S = (z_{j,k})_{(j,k) \in \Pi_2(S)}$, where

$$\Pi_2(S) = \{(j,k) : j < k; j, k \in S\} \quad (6)$$

is the set of all pairs in $S$ where the first entry has a lower index than the second one. The matrix $A_S$ has $d$ rows and $K(K-1)/2$ columns. Equipped with these matrices, the D-optimal design (Pukelsheim, 2006) is solved as

$$\pi_* = \arg\max_{\pi \in \Delta^\mathcal{S}} g(\pi), \quad \text{where} \quad (7)$$

$$g(\pi) = \log \det(V_\pi), \quad V_\pi = \sum_{S \in \mathcal{S}} \pi(S) A_S A_S^\top,$$

$\pi \in \Delta^\mathcal{S}$ is a probability distribution over the subsets in $\mathcal{S}$, $\Delta^\mathcal{S}$ is the simplex of all such distributions, and $\pi(S)$ denotes the probability of choosing the subset $S$ under $\pi$. Note that we could have indexed $\pi$ by an integer defined through a bijective mapping $\mathcal{C} : \mathcal{S} \to [\binom{N}{K}]$ from the subsets to natural numbers. We do not do this to simplify presentation. The problem (7) is concave since $\log \det$ is concave for PSD matrices and all $\pi(S) A_S A_S^\top$ are PSD by design. Moreover, its solution is sparse (Kiefer & Wolfowitz, 1960). Therefore, fast convex optimization methods, such as the Frank-Wolfe method, can be used to solve it. After $\pi_*$ is computed, the human feedback is collected on sampled subsets $S_\ell \sim \pi_*$.

We go beyond prior works in two aspects. First, we learn to rank $N \geq K$ items from $K$-way feedback. This can be done by combining (7) and (3), and we analyze this approach in

Section 3.3. Second, (7) cannot be solved efficiently because $\mathcal{S}$ is exponentially large in $K$. To address this challenge, we propose a randomized Frank-Wolfe algorithm in Section 5.

### 3.3. Generalization Analysis

In this section, we analyze active learning by the D-optimal design for ranking $N \geq K$ items from $K$-way feedback. We start with proving that the differences in estimated item relevance under $\hat{\theta}$ converge to those under $\theta_*$ as the sample size $T$ increases. The proof is under two assumptions that are not essential and only avoid rounding. We also assume that the parameter and feature vectors are bounded, as in Mukherjee et al. (2024), which allows to derive $C$ in (5).

**Assumption 1.** For all $k \in [N]$, $\|x_k\|_2 \leq 1$. In addition, $\|\theta_*\|_2 \leq 1$ and $\|\hat{\theta}\|_2 \leq 1$.

We can ensure that $\|\hat{\theta}\|_2 \leq 1$ when optimizing (3) because $\Theta = \left\{ \theta \in \mathbb{R}^d : \|\theta\|_2 \leq 1 \right\}$ is a convex set and (3) is a convex function. As a result, this can be done using gradient descent with a projection step to $\Theta$.

Our first claim is proved below.

**Proposition 1.** Let $K$ be even and $N/K$ be an integer. Let the feedback be collected according to $\pi_*$ in (7). Then with probability at least $1 - \delta$,

$$\sum_{j=1}^{N} \sum_{k=j+1}^{N} \left( z_{j,k}^\top (\theta_* - \hat{\theta}) \right)^2 = \tilde{O} \left( \frac{N^2 K^4 d^2 \log(1/\delta)}{T} \right),$$

where $z_{j,k} = x_j - x_k$.

*Proof.* We build on Theorem 5 of Mukherjee et al. (2024), which says that for any collection $\mathcal{S}$ of $K$-subsets,

$$\max_{S \in \mathcal{S}} \sum_{(j,k) \in \Pi_2(S)} \left( z_{j,k}^\top (\theta_* - \hat{\theta}) \right)^2 = \tilde{O} \left( \frac{K^6 d^2 \log(1/\delta)}{T} \right)$$

holds with probability at least $1 - \delta$. To reuse this result, we design a collection $\mathcal{C}$ such that any item pair appears in at least one $S \in \mathcal{C}$. Then any $(z_{j,k}^\top (\theta_* - \hat{\theta}))^2$ would be bounded. Let $\mathcal{I} = \{I_\ell\}_{\ell \in [2N/K]}$ be a partition of $[N]$ into sets with consecutive item indices, each of size $K/2$. Then $\mathcal{C}$ can be designed as follows. The first $2N/K - 1$ sets in $\mathcal{C}$ contain items $I_1$ combined with any other set $\mathcal{I} \setminus I_1$, the next $2N/K - 1$ sets in $\mathcal{C}$ contain items $I_2$ combined with any other set $\mathcal{I} \setminus I_2$, and so on. Clearly, the size of $\mathcal{C}$ is at most $4N^2/K^2$ and all $(z_{j,k}^\top (\theta_* - \hat{\theta}))^2$ are covered. Thus

$$\sum_{j=1}^{N} \sum_{k=j+1}^{N} \left( z_{j,k}^\top (\theta_* - \hat{\theta}) \right)^2 \leq \frac{4N^2}{K^2} \tilde{O} \left( \frac{K^6 d^2 \log(1/\delta)}{T} \right).$$

This concludes the proof. $\square$

The bound in Proposition 1 is $O(d^2/T)$, which is standard for a squared prediction error in linear models with $d$ parameters and sample size $T$. The dependencies on $N^2$, $K^4$, and $\log(1/\delta)$ are due to bounding predictions errors of $O(N^2)$ item pairs, from relative $K$-way feedback with probability at least $1 - \delta$.

Now we derive an upper bound on the ranking loss $R(T)$.

**Proposition 2.** *Let the feedback be collected according to $\pi_*$ in (7). Then the ranking loss is bounded as*

$$R(T) \leq \frac{4}{N(N-1)} \sum_{j=1}^{N} \sum_{k=j+1}^{N} \exp \left[ -\frac{(z_{j,k}^\top \theta_*)^2}{CK^4 d} + d \right],$$

*where $C > 0$ is a universal constant from the concentration analysis in Lemma 9 in Mukherjee et al. (2024).*

*Proof.* We build on Theorem 6 of Mukherjee et al. (2024). Specifically, the key step in their proof is that

$$\mathbb{P} \left( x_j^\top \hat{\theta} \leq x_k^\top \hat{\theta} \right) \leq \exp[-(z_{j,k}^\top \theta_*)^2 T/(CK^4 d) + d]$$

holds for any set of $K$ items $S$ and items $(j, k) \in \Pi_2(S)$ in it. Our claim follows from noting that $\mathbb{P}(\hat{\sigma}(j) > \hat{\sigma}(k)) = \mathbb{P} \left( x_j^\top \hat{\theta} \leq x_k^\top \hat{\theta} \right)$ and then applying the above bound. $\square$

Proposition 2 says that the ranking loss decreases exponentially with sample size $T$ and squared gaps; and increases with the number of features $d$ and $K$. The dependence on $T$, gaps, and $d$ is similar to prior works on fixed-budget best-arm identification in GLMs (Theorem 2 of Azizi et al. (2022)). Therefore, although we do not prove a lower bound, our bound is likely near-optimal.

### 3.4. Comparison to Clustering-based Approaches

In this section we argue that optimal design works better for data selection than clustering-based methods such as DBSCAN and $m$-medoids (see Appendix D.1 for details). As we saw in Sections 3.2 and 3.3, optimal design aims to minimize the worst-case estimation error along any direction (4). The $G$-Optimal Design objective directly targets minimizing this upper-bound in the worst case through the objective $h(\pi) = \max_z \|z\|_{M^{-1}}$, where $M$ depends on $\pi$. Then, by the Kiefer-Wolfowitz theorem (Kiefer & Wolfowitz, 1960; Mukherjee et al., 2024), maximizing $h(\pi)$ is equivalent to minimizing our D-Optimal Design objective $g(\pi) = \log \det(V_\pi)$ (7).

In contrast, clustering-based data selection methods aim to maximize data coverage by first clustering to minimize the total intra-cluster variance and then selecting centers/medoids from these clusters. Thus, when the original data distribution is unbalanced, clustering methods can miss potentially useful directions with fewer data points.

---

**Algorithm 1** Frank-Wolfe method for solving the D-optimal design in (7).

---

**Input:** #steps $T_{\mathrm{od}}$, initial iterate $\pi_0 \in \Delta^{\mathcal{S}}$

1 **for** $t = 0, 1, \ldots, T_{\mathrm{od}-1}$ **do**
2     Compute gradient: $G_t = \nabla_\pi g(\pi_t)$
3     LMO: $\widehat{\pi}_t \in \arg\max_{\pi \in \Delta^{\mathcal{S}}} \langle G_t, \pi \rangle$
4     Line search: $\alpha_t \in \arg\max_{\alpha \in [0,1]} g((1-\alpha) \cdot \pi_t + \alpha \cdot \widehat{\pi}_t)$
5     Iterate update: $\pi_{t+1} = (1-\alpha_t) \cdot \pi_t + \alpha_t \cdot \widehat{\pi}_t$
6 **return** $\pi_{T_{\mathrm{od}}}$

---

## 4. Frank-Wolfe Method for Optimal Design

The *Frank-Wolfe (FW)* method has been traditionally utilized as a scalable algorithm for solving D-optimal designs (Khachiyan, 1996; Zhao & Freund, 2023). When applied to (7), we obtain Algorithm 1. Each iteration of the method comprises three steps. First, we compute the gradient $G_t$ at the current iterate (Line 2 of Algorithm 1). Second, we find the distribution $\widehat{\pi}_t$ that maximizes the linear functional defined by $G_t$ (Line 3 of Algorithm 1) using a *linear maximization oracle (LMO)*. Finally, we update the iterate with a convex combination of the current iterate $\pi_t$ and $\widehat{\pi}_t$ with a step size $\alpha_t$, which is set to maximize $g((1-\alpha) \cdot \pi_t + \alpha \cdot \widehat{\pi}_t)$ (Line 4 of Algorithm 1) using *linear search*. It is known that this method converges to the maximizer of (7) (Zhao & Freund, 2023).

The FW method is efficient when the LMO can be implemented efficiently. In our problem (7) on simplex $\Delta^{\mathcal{S}}$, the LMO can be written

$$\widehat{\pi}_t = e_{S_t}, \text{ where} \tag{8}$$
$$S_t = \arg\max_{S \in \mathcal{S}} G_t(S), \quad e_{S_t}(S) = \mathbf{1}\{S = S_t\}.$$

Simply put, $\widehat{\pi}_t$ is a distribution where all probability mass is put on set $S_t$, which has the largest corresponding partial derivative $\max_{S \in \mathcal{S}} G_t(S)$. Note that this implies that line search with $e_{S_t}$ can only increase the number of non-zero elements in the iterate $\widehat{\pi}_t$ by at most one. So, after $t$ steps, the number of non-zero elements in $\widehat{\pi}_t$ is at most $t$ plus the number of non-zero elements in $\pi_0$. This is one reason why the FW method is often preferred over gradient descent.

While the LMO in (8) seems simple, it requires computing the gradient with respect an $\binom{N}{K}$-dimensional vector and finding its maximum entry. When $N \gg K$, this means that the computational complexity is exponential in $K$. As an example, in our experiments (Section 6), the FW method runs out of memory when $N = 100$ and $K = 10$ because the full gradient is a $10^{13}$-dimensional vector. Even under conservative estimates, this requires 68 TB of RAM for a float32 precision, which is out of reach for most practitioners. We tried parallelization and could compute $10^5$ partial

---

**Algorithm 2** `DopeWolfe`: A randomized FW method for solving the D-optimal design in (7).

---

**Input:** sampling size $R$, #steps $T_{\mathrm{od}}$, initial subset $S_0 \in \mathcal{S}$, regularization $\gamma$, and line-search tolerance $\alpha_{\mathrm{tol}}$

1 Let $\mathcal{S}$ be the collection of all $K$-subsets of $[N]$
2 Set $z_{j,k} = x_j - x_k, \forall (j,k) \in \Pi_2([N])$ and $\pi_0 = e_{S_0}$
    // Update iterate (Algorithm 4)
3 Set $V_0^{\mathrm{inv}} = \texttt{UpdateInverse}(\mathbf{I}_{d \times d}, A_{S_0}/(1-\gamma), 1-\gamma)$
4 **for** $t = 0, 1, \ldots, T_{\mathrm{od}-1}$ **do**
    // Randomized LMO
5     Sample $\mathcal{R}_t \sim \mathrm{Uniform}(\{\mathcal{R} \subseteq \mathcal{S} \mid |\mathcal{R}| = R\})$
6     $\{G_t(S)\} = \texttt{PartialGrad}(N, \mathcal{R}_t, V_t^{\mathrm{inv}}, \{z_{j,k}\})$
7     $S_t \in \arg\max_{S \in \mathcal{R}_t} G_t(S)$
    // Line search (Algorithm 4)
8     $\alpha_t = \texttt{GoldenSearch}(V_t^{\mathrm{inv}}, A_{S_t}, \alpha_{\mathrm{tol}})$
    // Update iterate (Algorithm 4)
9     $\pi_{t+1} = (1-\alpha_t) \cdot \pi_t + \alpha_t \cdot e_{S_t}$
10    $V_{t+1}^{\mathrm{inv}} = \texttt{UpdateInverse}(V_t^{\mathrm{inv}}, A_{S_t}, \alpha_t)$
11 **return** $\pi_{T_{\mathrm{od}}}$
12 **Sub-routine** `PartialGrad`$(N, \mathcal{R}, V^{\mathrm{inv}}, \{z_{j,k}\})$
13    **for** $(j,k) \in \Pi_2([N])$ **do**
     // Pair gradients
14     $D_{j,k} \leftarrow z_{j,k}^\top V^{\mathrm{inv}} z_{j,k}$
15    **for** $S \in \mathcal{R}_t$ **do**
16     $G(S) = \sum_{(j,k) \in \Pi_2(S)} D_{j,k}$
17    **return** $\{G(S)\}_{S \in \mathcal{R}_t}$

---

gradients at a time. This reduced the per-iteration computational complexity of the FW method to $10^8 = 10^{13}/10^5$. This is again impractical.

The line search can also be computationally costly because it requires objective calculations at many feasible iterates and each such calculation involves computing the $\log\det$ of a $d \times d$ matrix, which requires $O(d^3)$ operations in most practical implementations. In Section 5, we address these and other limitations through a new algorithm `DopeWolfe`, which has a better per-iteration complexity in terms of $N$, $K$, and $d$. Zhao (2023) proposed an away step variant of the FW method that has a superior empirical convergence rate for solving optimal designs. This method would face the same computation challenge as the FW method. We discuss this in detail in Appendix A.

## 5. `DopeWolfe`: Randomized Frank-Wolfe Method for Optimal Design

We propose and analyze our algorithm for solving the D-optimal design in (7) next. We call it `DopeWolfe` and show its pseudo-code in Algorithm 2. The pseudo-code for some sub-routines is in Appendix. `DopeWolfe` is a fast randomized variant of the FW method that incorporates compu-

tationally efficient memoization, and low-rank and sparse operations. In the rest of the section, we describe the building blocks of the algorithm and show how they address the scaling concerns identified in Section 4.

### 5.1. Randomized LMO and Cached Derivatives

The computational complexity of the LMO in Section 4 is $O(\binom{N}{K})$ because the maximization step is over an exponentially large $\mathcal{S}$ in (8). We utilize a randomized variant of the FW method (Kerdreux et al., 2018) to reduce it to $O(R)$ by restricting the maximization to an $R$-subset $\mathcal{R}_t$ chosen uniformly at random from $\mathcal{S}$ (Lines 5 and 7 of Algorithm 2),

$$S_t \in \arg\max_{S \in \mathcal{R}_t} G_t(S), \text{ where}$$
$$\mathcal{R}_t \sim \text{Uniform}(\{\mathcal{R} \subseteq \mathcal{S} \mid |\mathcal{R}| = R\}).$$

This implies that we only need to compute $R$ partial gradients $G_t(S) = \frac{\partial g(\pi_t)}{\partial \pi_t(S)}$ for the subsets $S \in \mathcal{R}_t$.

Next we show how the partial gradients can be computed efficiently. Recall the definitions of $g(\pi)$ and $V_\pi$ from (7). Since the gradient of $\log \det(U)$ is $U^{-1}$ for a PSD matrix $U$, the partial derivative at any $S$ can be written as

$$\frac{\partial g(\pi)}{\partial \pi(S)} = \langle A_S A_S^\top, V_\pi^{-1} \rangle = \sum_{(j,k) \in \Pi_2(S)} \underbrace{z_{j,k}^\top V_\pi^{-1} z_{j,k}}_{D_{j,k}}, \quad (9)$$

where $D_{j,k} = z_{j,k}^\top V_\pi^{-1} z_{j,k}$. Simply put, all partial derivatives are sums of $\binom{K}{2}$ terms chosen from the $\binom{N}{2}$ terms in $\{D_{j,k}\}_{(j,k) \in \Pi_2([N])}$. In DopeWolfe, we leverage this structure in sub-routine PartialGrad, which first computes and caches all $D_{j,k}$ (Line 14 in Algorithm 2), and then combines them to obtain partial derivatives for all $S \in \mathcal{R}_t$ (Line 16 in Algorithm 2). Assuming that $V_\pi^{-1}$ is known, this reduces the overall LMO complexity from $O(\binom{N}{K}\binom{K}{2}d^2)$ in Section 4 to $O(\binom{N}{2}d^2 + R\binom{K}{2})$. We parallelize these computations in our code.

### 5.2. Line Search with Low-Rank and Sparse Updates

We noted earlier that line search (Line 4 of Algorithm 1) is computationally costly. In DopeWolfe, we replace it with sub-routine GoldenSearch (Line 8 of Algorithm 2), which is provided in Algorithm 4 in Appendix. This sub-routine solves a one-dimensional unimodal maximization problem using the golden-section search (Kiefer, 1953). The golden-section search reduces the search space by a multiplicative factor of $\varphi$, known as the golden ratio, per iteration. As a result, it is guaranteed to return the maximizer up to an error $\alpha_{\text{tol}}$ after $1 + \log_\varphi(\alpha_{\text{tol}})$ iterations.

We also noted earlier that a naive computation of the objective value in line search has a complexity of $O(d^3)$. Here

we note that the update can be written as

$$(1 - \alpha)V_\pi + \alpha A_S A_S^\top. \quad (10)$$

Let $r = \binom{K}{2} \ll d$. Then $\alpha A_S A_S^\top$ is a rank-$r$ matrix since $A_S \in \mathbb{R}^{d \times r}$. Assuming access to $V_\pi^{-1}$, the log-determinant of (10) can be computed using that of an $r \times r$ matrix. This update is derived in Appendix C. This reduces the overall line search complexity to $O((r^3 + rd^2)\log_\varphi(\alpha_{\text{tol}}))$ from $O(d^3\alpha_{\text{tol}}^{-1})$ in Section 4.

Both the partial derivative and objective value computation require the inverse of $V_\pi$. DopeWolfe initializes the matrix by $V_0^{\text{inv}}$ (Line 3 of Algorithm 2) and then updates it (Line 10 of Algorithm 2) using UpdateInverse. The computational complexity of naively inverting a $d \times d$ matrix is $O(d^3)$. We note though that $V_\pi$ is updated by a low-rank matrix in (10). Therefore, we can compute the inverse of $V_\pi$ by using the inverse of an $r \times r$ matrix (Algorithm 4 in Appendix). As a result, the inverse computational complexity is reduced from $O(d^3)$ to $O(r^3 + rd^2)$.

Finally, we address how DopeWolfe maintains and stores the iterate $\pi_t$ in exponentially-many dimensions $\binom{N}{K}$. When $N \gg K > 1$, it becomes both computationally and space costly. Since DopeWolfe is initialized with a one-hot vector and its updates are sparse (Lines 2 and 9 of Algorithm 2), we store $\pi_t$ as a sparse vector and update it using sparse operations. This reduces the complexity of maintaining $\pi_t$ to its number of non-zero elements, which is at most $t + 1$ after $t$ iterations. We discuss this in detail in Appendix C.

### 5.3. Convergence Rate of Randomized FW Method

We analyze the convergence rate of DopeWolfe next. Since DopeWolfe is a randomized FW method, we know that it converges to the maximizer of (7) if $g$ was Lipschitz smooth (Kerdreux et al., 2018). However, $g$ does not satisfy this condition because of the logarithm in its definition. To provide guarantees, we prove a more general result, that a randomized FW method (Algorithm 3 in Appendix) converges for problems of the form

$$\min_{y \in \mathcal{Y}} F(y) = \min_{y \in \mathcal{Y}} f(\mathcal{A}(y)), \quad (11)$$

where $\mathcal{Y}$ is the convex hull of set $\mathcal{V}$ of $\widetilde{N}$ vectors, $\mathcal{A} : \mathcal{Y} \to \mathcal{K}$ is a linear operator, $\mathcal{K}$ is a regular cone, and $f : \text{int}(\mathcal{K}) \to \mathbb{R}$ is a logarithmically-homogeneous self-concordant barrier (LHSCB) (Zhao & Freund, 2023).

**Theorem 3.** *If the initial iterate $y_0$ maps to domain of $f$, i.e. $\mathcal{A}(y_0) \in \text{int}(\mathcal{K})$, then Randomized FW method (Algorithm 3) which samples $R$ elements from $\mathcal{V}$ for computing the randomized LMO achieves an $\varepsilon$ sub-optimal solution to the problem (11) after $T_{\text{FW}} = O(\max(1, \widetilde{N}/R)\varepsilon^{-1})$ iterations, i.e. $\mathbb{E}[F(y_{T_{\text{FW}}})] - \min_{y \in \mathcal{Y}} F(y) \le \varepsilon$.*

This result extends the convergence analysis of a randomized FW method for Lipschitz smooth objectives to problems of the form (11). As a corollary of Theorem 3, we provide the convergence rate for `DopeWolfe`.

**Corollary 4.** *For full-rank $V_{\pi_0}$, $\gamma = 0$, and small enough $\alpha_{\mathrm{tol}}$, `DopeWolfe` (Algorithm 2) outputs an $\varepsilon$ sub-optimal solution to the D-optimal design problem (7) after $T_{\mathrm{od}} = O(\max(1, \binom{N}{K}/R)\varepsilon^{-1})$ iterations, i.e. $\max_{\pi \in \Delta^s} g(\pi) - \mathbb{E}[g(\pi_{T_{\mathrm{od}}})] \leq \varepsilon$.*

We provide the proofs of these statements in Appendix B. Note that Corollary 4 needs $V_{\pi_0}$ to be full-rank; otherwise the objective might not be well-defined. In practice, this can be guaranteed by regularizing the matrix as $V_{\pi_0} + \gamma \mathbf{I}_{d \times d}$ with a small $\gamma > 0$. Finally, aggregating the complexities of all steps of `DopeWolfe`, we observe that its per-iteration complexity is $O(\binom{N}{2}d^2 + R\binom{K}{2} + (r^3 + rd^2)\log_\varphi(\alpha_{\mathrm{tol}}) + T_{\mathrm{od}})$, eliminating any $\binom{N}{K}$ and $d^3$ dependence.

Note that `DopeWolfe` without a randomized LMO would have the per-iteration complexity of $O(\binom{N}{2}d^2 + \binom{N}{K}\binom{K}{2} + (r^3 + rd^2)\log_\varphi(\alpha_{\mathrm{tol}}) + T_{\mathrm{od}})$ and the iteration complexity of $O(\varepsilon^{-1})$ (Zhao & Freund, 2023). As a result, in theory, the randomized LMO has a better per-iteration complexity while increasing the worst-case iteration complexity to $O(\binom{N}{K}/(R\varepsilon))$. Note that this is desirable because the FW method is impractical even for small instances of our problems, as discussed in Section 4. We do not observe empirically (Section 6) that the convergence rate of `DopeWolfe` is exponentially worse, as bounded in Corollary 4. Similar observations have been made in prior works, where the empirical convergence rate of the randomized FW method is much better than the theoretical upper bound when applied to Lipschitz smooth problems (Kerdreux et al., 2018, Figure 1). Therefore, we conjecture that Theorem 3 is not tight for most problems and defer further study to future work.

## 6. Experiments

We evaluate the performance of `DopeWolfe` in three experiments. In Section 6.1, we compare it to `Dope` (Mukherjee et al., 2024) on the same datasets and in the same setting. This is the closest related work. The main difference in our algorithm is that it can be applied to larger problems. We evaluate `DopeWolfe` on such problems with synthetic feedback in Section 6.2 and with real feedback in Section 6.3. The main evaluation metric is the *mean ranking loss* in (2). We also report NDCG in Appendix.

### 6.1. Comparison to Dope

In the first experiment, we compare to `Dope` (Mukherjee et al., 2024) on both Anthropic (Zhu et al., 2023a) and Nectar (Bai et al., 2022) datasets in their work. We implement `Dope` using Algorithm 1. This is a major computational im-

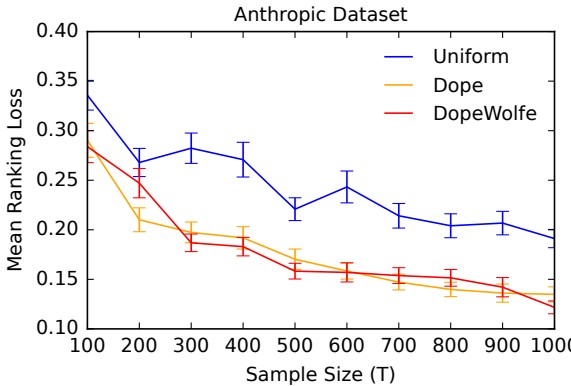

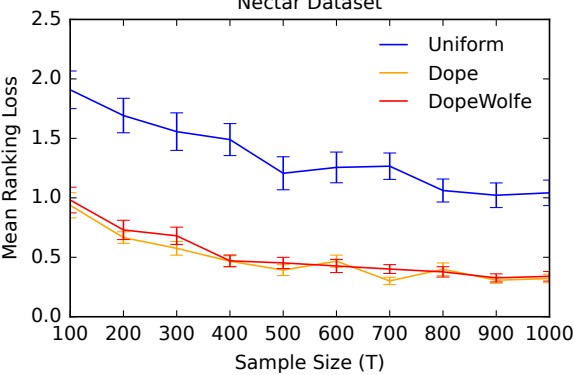

*Figure 1.* Mean ranking loss of `DopeWolfe`, `Dope`, and `Uniform` on the Antropic and Nectar problems in Mukherjee et al. (2024).

provement over the original implementation of Mukherjee et al. (2024), which solves the linear maximization problem in (8) using linear programming. Moreover, we replace the line search in Algorithm 1 with the golden-section search in Algorithm 2. We also compare to a baseline that chooses subsets of items uniformly at random. While simple, it is known to be competitive in real-world problems where feature vectors may cover the feature space close to uniformly (Ash et al., 2019; Yuan et al., 2020; Ash et al., 2021; Ren et al., 2021). We call it `Uniform`. `DopeWolfe` is run with 10% of randomly sampled subsets of items.

We vary the sample size from $T = 100$ to $T = 1\,000$, and report our results in Figure 1. We observe that the ranking losses of `DopeWolfe` and `Dope` are similar, while those of `Uniform` are much worse. We note a major difference in the run times of `DopeWolfe` and `Dope`. Although we implement `Dope` more efficiently than Mukherjee et al. (2024), `DopeWolfe` is more than three times faster on the Anthropic dataset (6.184 seconds on average versus 20.147) and more than two times faster on the Nectar dataset (0.411 seconds on average versus 0.960). This is despite the fact that our method is designed for much larger problems.

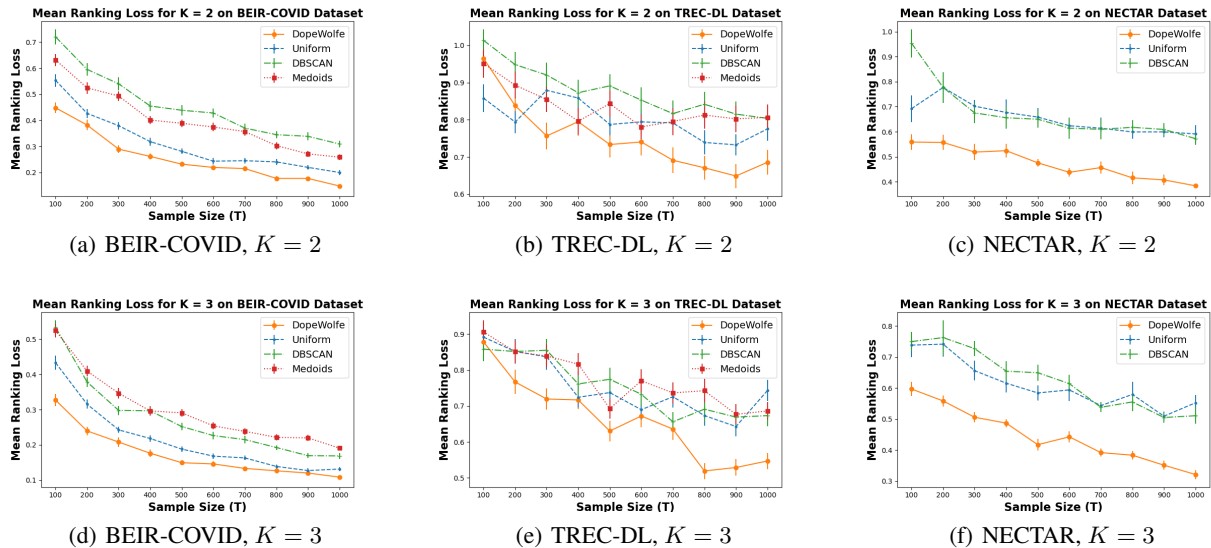

*Figure 2.* Mean ranking loss on three datasets with synthetic feedback. `Medoids` fails on the Nectar dataset due to excessive memory requirements, 2 TB and 8 TB of RAM, respectively.

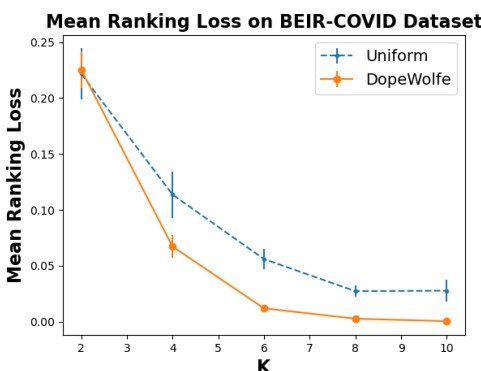

(a) Ranking loss on BEIR-COVID with real feedback.

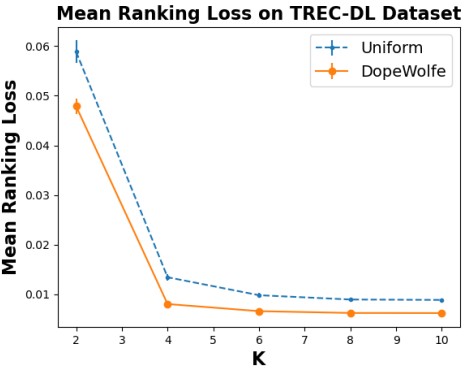

(b) Ranking loss on TREC-DL with real feedback.

*Figure 3.* Mean ranking loss on two datasets with real feedback.

## 6.2. Synthetic Feedback

Our second experiment is on problems where `Dope` cannot be implemented efficiently. We experiment with two text retrieval datasets, BEIR-COVID[1] and TREC Deep Learning (TREC-DL)[2], and Nectar (Bai et al., 2022) dataset. In the first two datasets, we experiment with single lists of $N = 100$ items. In the Nectar dataset, we choose $30\,000$ lists, each with 7 items, and learn to rank of these lists simultaneously, as described in the last paragraph in Section 2. Even for $K = 2$, this means $30\,000 \cdot \binom{7}{2} = 630\,000$ subsets of items to choose from, which is two orders of magnitude more than in Section 6.1.

The feature vectors of the items are their 100-dimensional BERT embeddings. The parameters of the Plackett-Luce model are learned from data and the feedback is sampled according to (1) with those parameters. We consider three baselines: `Uniform`, which was already introduced in Section 6.1; `DBSCAN` (Ester et al., 1996), which is a clustering baseline; and `Medoids`, which is another clustering baseline. We describe the baselines in more detail in Appendix D.1. All results are averaged over 100 random runs.

We vary the sample size from $T = 100$ to $T = 1\,000$, and report results for $K \in \{2, 3\}$ in Figure 2. We observe the following trends. First, `DopeWolfe` has the lowest ranking loss. Some improvements are of an order of magnitude. In particular, for $K = 2$ on the Nectar dataset, the ranking loss of `DopeWolfe` at $T = 100$ is lower than that of all

---

[1]https://huggingface.co/datasets/BeIR/trec-covid
[2]https://microsoft.github.io/msmarco/TREC-Deep-Learning

baselines at $T = 1\,000$. Second, both DBSCAN and Medoids perform relatively poorly because they often do not even outperform Uniform. We also note that the ranking losses of all methods decrease as $T$ and $K$ increase. The former is expected because the methods benefit from more feedback. The latter is also expected because each unit of feedback contains more information, due to ranking more items. For completeness, we report NDCG in Figure 4 (Appendix D.1) and results for $K = 4$ in Figure 5 (Appendix D.1). We also compare the run times of DopeWolfe and DBSCAN in Appendix D.3.

### 6.3. Real Feedback

The third experiment is a more realistic variant of the experiment in Section 6.2. There are three main differences. First, we use real feedback instead of simulated feedback. The real feedback is more challenging since the optimized model may be and likely is misspecified. Second, we use more representative 1024-dimensional BGE-M3 embeddings (Chen et al., 2024) to ensure that the features are sufficiently rich to learn the true ranking order under real feedback. Finally, we explore a wider range of feedback with $K \in \{2, 4, \ldots, 10\}$. Note that large $N$ and moderate $K$ make this experiment an extremely large-scale ranking problem. Particularly, for $N = 100$ and $K = 10$, there are 17 trillion subsets of items for eliciting feedback. Therefore, it is essentially impossible to run prior optimal design methods (Mukherjee et al., 2024) or the Frank-Wolfe method (Zhao & Freund, 2023). We omit DBSCAN and Medoids baselines due to the same scale issues. We provide more details in Appendix D.2.

We plot all results as a function of $K$ and report them in Figure 3. We observe the following trends. First, DopeWolfe has the lowest ranking loss in all settings. Second, the ranking losses of both methods decrease as $K$ increases. This is expected because each unit of feedback contains more information, due to ranking more items. While Figures 2 and 3 are not comparable, they can be compared by looking at relative differences in the ranking losses of Uniform and DopeWolfe. This is consistently $20\%$ in Figure 3 and similar to many relative differences in Figure 2. We conclude the despite the more challenging settings, the improvements due to DopeWolfe can be of a similar magnitude. For completeness, we report NDCG in Figure 6 (Appendix D.2).

## 7. Conclusions

We study learning to rank $N$ items from $K$-way ranking feedback under a limited feedback budget. We develop an D-optimal design framework for eliciting $K$-way feedback by generalizing Mukherjee et al. (2024) and prove that it is statistically efficient in minimizing the ranking loss. We also show that the classic methods for solving this problem are infeasible in our setting because they optimize over $O\left(\binom{N}{K}\right)$

variables, exponentially many in $K$.

To address this challenge, we propose DopeWolfe for solving our optimal design problem. DopeWolfe is a randomized FW method that uses memoization and sparse updates to improve the per-iteration complexity of the FW method from $O\left(\binom{N}{K}\right)$ to $O(N^2 + K^2)$. Furthermore, we provide a convergence analysis for DopeWolfe by generalizing the analysis of the randomized FW method beyond Lipschitz smoothness to a general class of logarithmically homogeneous self-concordant barrier objectives. Finally, we empirically demonstrate the computational and statistical efficiency of our framework on both synthetic and real-world datasets, with large $N$ and moderate $K$.

**Limitations.** Our work can be viewed as learning to rank $N$ items from relative $K$-way feedback. A natural question to ask is if this can be done using absolute feedback. In our opinion, absolute feedback has three major shortcomings. First, annotators often exhibit variable calibration when assigning absolute scores, making cross-annotator aggregation difficult. Second, comparing items is typically more natural and faster for humans. Finally, often we only have access to proxy signals (ad clicks or video watch time), which have an approximately monotonic relation to the real preference. A relative comparison requires only ordering rather than judging the precise value, mitigating calibration and proxy issues. Ultimately, whether the absolute or relative feedback is used is a design choice, and we believe that both should be studied.

We have an offline active learning problem, in a sense that DopeWolfe selects $K$-subsets of items to elicit feedback on before receiving any feedback. The most natural way of making it online would be to update item embeddings after receiving feedback.

**Future work.** Our work can be extended in multiple directions. First, we only studied one particular human feedback model, $K$-way ranking, popularized by RLHF (Ouyang et al., 2022) and DPO (Rafailov et al., 2023). Second, the $O(d^2)$ dependence in Proposition 1 is suboptimal. It can be reduced to $O(d)$ by avoiding the Cauchy-Schwarz inequality in (4). Finally, we want to conduct more realistic experiments on high-dimensional LLM embeddings used in the reward modeling phase of RLHF (Sun et al., 2023).

## Impact Statement

This paper presents work whose goal is to advance the field of Machine Learning. There are many potential societal consequences of our work, none which we feel must be specifically highlighted here.

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

# A. Related Work

Three recent papers studied a similar setting. Mehta et al. (2023) and Das et al. (2024) learn to rank pairs of items from pairwise feedback. Mukherjee et al. (2024) learn to rank lists of $K$ items from $K$-way feedback in (1). We propose a general framework for learning to rank $N$ items from $K$-way feedback, where $K \geq 2$ and $N \geq K$. Therefore, our setting is more general than in prior works. Our setting is also related to other bandit settings as follows. Due to the sample budget, it is reminiscent of fixed-budget *best arm identification (BAI)* (Bubeck et al., 2009; Audibert et al., 2010; Azizi et al., 2022; Yang & Tan, 2022). The main difference is that we do not want to identify the best arm. We want to estimate the mean rewards of $N$ items to sort them. Online learning to rank has also been studied extensively (Radlinski et al., 2008; Kveton et al., 2015; Zong et al., 2016; Li et al., 2016; Lagree et al., 2016; Hiranandani et al., 2019). We do not minimize cumulative regret or try to identify the best arm.

We discuss related work on Frank-Wolfe (FW) methods next. The use of FW method for solving standard optimal design problem (for linear and logistic regression) is not new. In fact, it can be traced even up to 1990s (Khachiyan, 1996). Ahipaşaoğlu (2015) presents two first-order Frank-Wolfe algorithms with rigorous time-complexity analyses for the A-optimal experimental design. Hendrych et al. (2023) optimize an experimental design by the Frank-Wolfe algorithm within a mixed-integer convex optimization framework. Zhao & Freund (2023) introduce a generalized Frank-Wolfe method for composite optimization with a logarithmically-homogeneous self-concordant barrier (LHSCB). None of these works focus on ranking problems. Please see Zhao & Freund (2023) for a comprehensive literature review. Recently Zhao (2023) proposed and analyzed Away-step variant of the FW (AFW) method for LHSCB problems. Just like in the case of Lipschitz smooth problems, AFW achieves better empirical performance than standard FW method. However, we would like to highlight that AFW is also infeasible for our problem since it has higher per-iteration cost than the standard FW method which is already infeasible. Additionally, their implementation (Zhao, 2023, Sec 6.2-6.3) nitializes $\pi_0$ with a uniform distribution, which can also be a practical challenge for us since $\pi$ is $O(N^K)$ dimensional. This is the reason we do not compare against this algorithm. Our randomized Frank-Wolfe algorithm addresses this specific computational challenge when solving for a D-optimal design for learning to rank with $O(\binom{N}{K})$ variables.

While other works have learning to rank with ranking feedback, motivated by learning preference models in RLHF (Rafailov et al., 2023; Kang et al., 2023; Casper et al., 2023), our work is unique in applying optimal designs to collect human feedback for learning to rank. Tennenholtz et al. (2024) show the benefits of using an optimal design of a state-dependent action set to improve an embedding-aligned guided language agent's efficiency. Additionally, Khetan & Oh (2016) and Pan et al. (2022) provide optimal rank-breaking estimators for efficient rank aggregation under different user homogeneity assumptions. Our work stands out by integrating optimal design with randomized Frank-Wolfe method in the context of ranking a large list of $N$ items through $K$-way ranking feedback.

In offline RL, the agent directly observes the past history of interactions. Note that these actions can be suboptimal and there can be issues of data coverage and distribution shifts. Therefore in recent years pessimism under offline RL has gained traction (Jin et al., 2021; Xu et al., 2022; Zanette, 2023). In contrast to these works, we study offline K-wise preference ranking under PL and BTL models for pure exploration setting. We do not use any pessimism but use optimal design (Pukelsheim, 2006; Fedorov, 2013) to ensure diversity among the data collected. If the action set is infinite, then approximately optimal designs can sometimes be found efficiently (Lattimore & Szepesvari, 2019). (Sun et al., 2023) investigated generative LLMs, such as ChatGPT and GPT-4, for relevance ranking in IR and found that properly instructed LLMs can deliver competitive, even superior results to state-of-the-art supervised methods on popular IR benchmarks. We note that our approach can further be used in such settings to enhance the performance.

There are also related works in the bandit community. Saha & Gopalan (2019a) proposed non-contextual algorithms. That is, they do not have access to features of the items and hence the relevance (Plackett-Luce score) of each item is estimated independently of other items. The paper studies two feedback models: the winner out of $K$ items and top-$m$ ranked items out of $K$ for $m \leq K$. The performance metric is a ranking loss. We differ from this work by having a contextual algorithm, which parameterizes the relevance of an item as a linear function of its feature. Szorenyi et al. (2015) also proposed non-contextual algorithms. The paper studies dueling feedback. The performance metrics are the best item and a ranking loss. We differ from this work by having a contextual algorithm and more general feedback than dueling ($K > 2$). Saha & Gopalan (2019b) also proposed non-contextual algorithms. The paper studies two feedback models: the winner out of $K$ items and top-$m$ ranked items out of $K$ for $m \leq K$. The performance metric is the best item. We differ from this work by having a contextual algorithm and different performance metric.

The most naive approach to solving our problem is uniform sampling. That is $\pi_{\text{uniform}}(S) = \binom{N}{K}^{-1} \ \forall S \in \mathcal{S}$. Its

convergence rate depends on the distribution of feature vectors and can be significantly worse than that of `DopeWolfe`. To see this, consider $K = 2$ and $N \gg d$. Let $x_1 = (1, 0, 0)$. For $k > 1$, let $x_k = (0, 1, 0)$ for even $k$ and $x_k = (0, 0, 1)$ for odd $k$. In this case, only $N - 1$ feature vector differences out of $N(N-1)/2$ have a non-zero entry in the first dimension. Therefore, the minimum eigenvalue of $V_\pi$ grows at the rate of $\Omega(T/N)$ in expectation, unlike $\Omega(T/d)$ in the D-optimal design. The implication for the bounds is that one $d$ in Proposition 1 is replaced with $N$ and the only $d$ in Proposition 2 is replaced with $N$.

## B. Theoretical convergence rate analyses

This section provides the theoretical convergence rate analysis of `DopeWolfe` (Algorithm 2). We do this by first proving a more general result for the broad class of LHSCB problems (11). Then we will provide the convergence rate of `DopeWolfe` as a corollary of this result.

### B.1. Generic Randomized FW method for solving the LHSCB problem

In this section we analyze the convergence of the generic randomize FW method (Algorithm 3) for minimizing composition of $\theta$-LHSCB functions and affine transformations. More precisely, we recall the studied optimization problem

$$\min_{y \in \mathcal{Y}} F(y) = \min_{y \in \mathcal{Y}} f(\mathcal{A}(y)), \tag{12}$$

where $\mathcal{Y}$ is the convex hull of a set $\mathcal{V}$ of $\widetilde{N}$ vectors ($\mathcal{Y} = \mathrm{conv}(\mathcal{V})$), $\mathcal{A} : \mathcal{Y} \to \mathcal{K}$ is a linear operator, $\mathcal{K}$ is a cone, and $f : \mathrm{int}(\mathcal{K}) \to \mathbb{R}$ is a $\theta$-*logarithmically-homogeneous self-concordant barrier (LHSCB)* (Zhao & Freund, 2023). Next we define the LHSCB function.

**Definition 5.** *We say that a function $f$ is $\theta$-logarithmically homogenous self-concordant barrier (LHSCB) if*

1. *$f : \mathcal{K} \to \mathbb{R}$ is a convex mapping with a regular cone $\mathcal{K} \subsetneq \mathbb{R}^m$ (closed, convex, pointed and with non-empty interior $\mathrm{int}(\mathcal{K})$) as its domain,*

2. *$f$ is $\theta$-logarithmically homogenous, i.e. $f(tu) = f(u) - \theta \log t$, for all $u \in \mathrm{int}(\mathcal{K})$ and $t > 0$ for some $\theta \geq 1$,*

3. *$f$ is self-concordant, i.e. $|D^3 f(u)[v, v, v]|^2 \leq 4\langle v, \nabla^2 f(u)v \rangle^3$ for all $u \in \mathrm{int}(\mathcal{K})$ and $v \in \mathbb{R}^m$, where $D^3 f(u)[v, v, v]$ is the third order derivative of $f$ at $u$ in the direction $v$ and $\nabla^2 f(u)$ is the Hessian of $f$ at $u$, and*

4. *$f$ is a barrier, i.e. $f(u_k) \to +\infty$ for any $(u_k)_{k \geq 0} \subseteq \mathrm{int}(\mathcal{K})$ and $u_k \to u \in \mathrm{bd}(\mathcal{K})$, where $\mathrm{bd}(\mathcal{K})$ is the boundary of $\mathcal{K}$.*

From the above assumption it is clear that problem (12) is a convex problem.

---

**Algorithm 3** Randomized Frank-Wolfe Method for LHSCB Problem (12)

---

**Input:** $y_0$ such that $\mathcal{A}(y_0) \in \mathrm{int}(\mathcal{K})$, sample size $R$

1 **for** $t = 0, 1, \ldots, T_{\mathrm{FW}}$ **do**

2 $\quad$ Sample $R \ (\leq \widetilde{N})$ subsets uniformly at random from $\mathcal{V}$: $\mathcal{R}_t \sim \mathrm{Uniform}(\{\mathcal{R} \subseteq \mathcal{S} \mid |\mathcal{R}| = R]\})$
$\quad\quad$ Compute $v_t = \underset{v \in \mathcal{R}_t}{\arg\min} \langle \nabla F(y_t), v \rangle$ `// Randomizes LMO`

3 $\quad$ Compute $\alpha_t = \underset{\alpha \in [0,1]}{\arg\min} F(x_t + \alpha(v_t - x_t))$ `// Line Search`

4 $\quad$ Update $x_{t+1} = x_t + \alpha_t(v_t - x_t)$

---

In Algorithm 3, we provide the generic randomize FW method for solving problem (12). Before providing its convergence rate we state a known property of LHSCB functions. It is know that LHSCB functions satisfy the following approximate second-order upper bound. We define, for a given positive semi-definite matrix $M$, the weighted norm $\|u\|_M = \sqrt{u^T M u}$.

**Proposition 6** ((Zhao & Freund, 2023; Nesterov, 2014)). *Let $f : \mathcal{K} \to \mathbb{R}$ be LHSCB. Then,*

$$f(u) \leq f(v) + \langle \nabla f(u), v - u \rangle + \omega(\|v - u\|_{\nabla^2 f(u)}), \quad \forall u \in \mathrm{int}(\mathcal{K}), v \in \mathcal{K} \tag{13}$$

*where*

$$\omega(a) = \begin{cases} -a - \ln(1-a), & a < 1 \\ +\infty, & 1 \le a. \end{cases} \tag{14}$$

Note that the $\omega$ is convex and non-negative in $(-\infty, 1)$.

Now we are ready to provide the convergence rate of Algorithm 3. For the sake of clarity, we restate Theorem 3 with more details below. This proof is inspired from the proof techniques developed in (Kerdreux et al., 2018; Zhao & Freund, 2023).

**Theorem 7.** *Let the initial iterate $y_0$ maps to domain of $f$, i.e. $\mathcal{A}(y_0) \in \text{int}(\mathcal{K})$, and $\delta_0 = F(y_0) - \min_{y \in \mathcal{Y}} F(y)$, then Randomized FW method (Algorithm 3) which uniformly samples $R$ elements from $\mathcal{V}$ for computing the randomized LMO at each step outputs an $\varepsilon$ sub-optimal solution to the $\theta$-LHSCB problem (11) in expectation, after*

$$T_{\text{FW}} = \left\lceil 5.3 \frac{\widetilde{N}}{R} (\delta_0 + \theta) \max\left\{ \ln\left(10.6 \frac{\widetilde{N}}{R} \delta_0\right), 0 \right\} \right\rceil + \left\lceil 12 \frac{\widetilde{N}}{R} \theta^2 \max\left\{ \frac{1}{\varepsilon} - \max\left\{ \frac{1}{\delta_0}, 10.6 \frac{\widetilde{N}}{R} \right\}, 0 \right\} \right\rceil, \tag{15}$$

*iterations, i.e. $\mathbb{E}[F(y_{T_{\text{FW}}})] - \min_{y \in \mathcal{Y}} F(y) \le \varepsilon$.*

*Proof.* First notice that since $\mathcal{A}$ is a linear mapping, the update rule in Algorithm 3 can be re-written as

$$v_t = \underset{v \in \mathcal{R}_t}{\arg\min} \langle \nabla F(y_t), v \rangle = \underset{v \in \mathcal{R}_t}{\arg\min} \langle \mathcal{A}^\top(\nabla f(\mathcal{A}(y_t))), v \rangle = \underset{v \in \mathcal{R}_t}{\arg\min} \langle \nabla f(\mathcal{A}(y_t)), \mathcal{A}(v) \rangle, \tag{16}$$

and

$$\alpha_t = \underset{\alpha \in [0,1]}{\arg\min} \ F((1-\alpha)y_t + \alpha v) = \underset{\alpha \in [0,1]}{\arg\min} \ f(\mathcal{A}((1-\alpha)y_t + \alpha v)). \tag{17}$$

Above also follows from the well-known affine invariance property of FW method (Jaggi, 2013). Next using Proposition 6 we get

$$\begin{aligned} F((1-\alpha)y_t + \alpha v_t) &= f(\mathcal{A}((1-\alpha)y_t + \alpha v_t)) \\ &\le f(\mathcal{A}(y_t)) + \langle \nabla f(\mathcal{A}(y_t)), \mathcal{A}((1-\alpha)y_t + \alpha v_t) \rangle + \omega(\|\mathcal{A}((1-\alpha)y_t + \alpha v_t)\|_{\nabla^2 f(\mathcal{A}(y_t))}) \\ &\le F(y_t) + \alpha \langle \nabla f(\mathcal{A}(y_t)), \mathcal{A}(v_t - y_t) \rangle + \omega(\alpha \|\mathcal{A}(v_t - y_t)\|_{\nabla^2 f(\mathcal{A}(y_t))}). \end{aligned}$$

Let $G_t = -\langle \nabla f(\mathcal{A}(y_t)), \mathcal{A}(v_t - y_t) \rangle$ and $D_t = \|\mathcal{A}(v_t - y_t)\|_{\nabla^2 f(\mathcal{A}(y_t))}$. We also assume that $v_t^* \in \arg\min_{v \in \mathcal{V}} \langle \nabla f(\mathcal{A}(y_t)), v \rangle$ is the output of the LMO if we run regular deterministic FW method (general version of Algorithm 1) at iteration $t$. Note that we can also define this LMO output as $v_t^* \in \arg\min_{v \in \mathcal{Y}} \langle \nabla f(\mathcal{A}(y_t)), v \rangle$ since $\mathcal{Y} = \text{conv}(\mathcal{V})$ and a minimizier of a linear functional on a closed convex polytope is one of its vertices. Using this definition of $v_t^*$ we also define $G_t^* = -\langle \nabla f(\mathcal{A}(y_t)), \mathcal{A}(v_t^* - y_t) \rangle$ and $D_t^* = \|\mathcal{A}(v_t^* - y_t)\|_{\nabla^2 f(\mathcal{A}(y_t))}$. Then taking the minimum w.r.t. $\alpha$ on both sides of the above inequality yields

$$F(y_{t+1}) = \min_{\alpha \in [0,1]} F((1-\alpha)y_t + \alpha v_t) \le F(y_t) + \min_{\alpha \in [0,1]} [-\alpha G_t + \omega(\alpha D_t)], \tag{18}$$

and by taking expectation conditioned on the randomness of $y_t$ we get

$$\mathbb{E}[F(y_{t+1}) \,|\, y_t] \le F(y_t) + \mathbb{E}[\min_{\alpha \in [0,1]} [-\alpha G_t + \omega(\alpha D_t)] \,|\, y_t]. \tag{19}$$

Notice that when $\alpha = 0$ we obviously have that $-\alpha G_t + \omega(\alpha D_t) = 0$ since $\omega(0) = 0$. Therefore, $\min_{\alpha \in [0,1]}[-\alpha G_t + \omega(\alpha D_t)] \le 0$. Note that, in contrast to regular FW method (Algorithm 1) where $G_t$ is always negative, for the worst case sampling of $\mathcal{R}_t$, the quantity $G_t$ can be positive. Therefore, choosing the $\alpha$ dynamically is necessary to ensure non-increasing objective values in randomized FW (Kerdreux et al., 2018). Next, we upper bound the above inequality as

$$\begin{aligned} \mathbb{E}[F(y_{t+1}) \,|\, y_t] &\le F(y_t) + (1 - \text{Prob}(v_t^* \in \mathcal{V}_t)) \cdot 0 + \text{Prob}(v_t^* \in \mathcal{V}_t) \min_{\alpha \in [0,1]} [-\alpha G_t^* + \omega(\alpha D_t^*)] \\ &= F(y_t) + (R/\widetilde{N}) \min_{\alpha \in [0,1]} [-\alpha G_t^* + \omega(\alpha D_t^*)]. \end{aligned} \tag{20}$$

Now notice that the above "descent lemma" inequality is similar to (Zhao & Freund, 2023, Inequality 2.7), except for the conditional expectation on the LHS and the discount factor $(R/\widetilde{N}) \le 1$ on second term which corresponds to the minimum decrease in objective value. Therefore, rest of the proof follow similar arguments as (**?**)Theorem 1]zhao2023analysis. □

This proves that randomized FW method convergence to the the minimizer of the LHSCB problem in $((\widetilde{N}/R)\varepsilon^{-1})$ steps. Compared to regular FW method, randomized version increases the iteration complexity by a factor of $\widetilde{N}/R$. We see similar increase in iteration complexity even for convex problems with bounded Lipschitz smoothness (Kerdreux et al., 2018). In the next section we specialize this result to provide a convergence rate guarantee for `DopeWolfe` (Algorithm 2).

### B.2. Theoretical analysis of `DopeWolfe`

Again for the sake of clarity we first restate Corollary 4 providing the iteration complexirt of `DopeWolfe` (Algorithm 4) with more details and then provide its proof.

**Corollary 8** (of Theorem 7). *Let $V_{\pi_0}$ be full-rank, $\gamma = 0$, $R \leq \binom{N}{K}$, $\alpha_{\text{tol}}$ be small enough and $\delta_0 = \max_{\pi \in \Delta^S} g(\pi) - g(\pi_0)$, then* `DopeWolfe` *(Algorithm 2) which uniformly samples $R$ subsets from $\mathcal{S}$ for computing the randomized LMO at each step, outputs an $\varepsilon$ sub-optimal solution to the D-optimal design problem (7) in expectation, after*

$$T_{\text{od}} = \left\lceil 5.3 \frac{\binom{N}{K}}{R}(\delta_0 + d) \max\left\{ \ln\left(10.6\frac{\binom{N}{K}}{R}\delta_0\right), 0\right\}\right\rceil + \left\lceil 12 \frac{\binom{N}{K}}{R}d^2 \max\left\{\frac{1}{\varepsilon} - \max\left\{\frac{1}{\delta_0}, 10.6\frac{\binom{N}{K}}{R}\right\}, 0\right\}\right\rceil \quad (21)$$

*iterations, i.e.* $\max_{\pi \in \Delta^S} g(\pi) - \mathbb{E}[g(\pi_{T_{\text{od}}})] \leq \varepsilon$.

*Proof.* Let $\mathbb{S}_+^d$ be the regular cone of all PSD matrices of dimension $d \times d$. Then we notice that the maximization problem in the D-optimal design (7) can be written as a special case of the minimization problem (12) by setting $F = -g$, $y = \pi$, $\mathcal{K} = \mathbb{S}_+^d$, $f = -\log \det : \mathbb{S}_+^d \to \mathbb{R}$, $\mathcal{A} = \sum_{S \in \mathcal{S}} \pi(S)A_S A_S^\top : \pi \to \mathbb{S}_+^d$, and $\mathcal{V} = \mathcal{S}$ ($\widetilde{N} = \binom{N}{K}$). It is known that $f = \log \det$ is an LHSCB function (Definition 5) with $\theta = d$ (Zhao & Freund, 2023). Moreover, $\mathcal{A}$ maps to $\mathbb{S}_+^d$ because $\pi(S)A_S A_S^\top$ is PSD for all $S \in \mathcal{S}$.

This equivalence implies that when $\alpha_{\text{tol}} \to 0$ and $\gamma = 0$, running `DopeWolfe` (Algorithm 2) is equivalent to running randomized FW method (Algorithm 3) for the above specialization. Please see Section 5 for description of how various sub-routines of Algorithm 2 implements varioius steps of Algorithm 3. Finally, we can satisfy the condition $y_0 = \text{int}(\mathcal{K})$ if $V_{\pi_0}$ is full rank since $V_{\pi_0} \in \text{int}(\mathbb{S}_+^d)$. Then the the iteration complexity of `DopeWolfe` directly follows from Theorem 7. $\square$

Note that above theorem requires that $\alpha_{\text{tol}}$ is small enough, but since `GoldenSearch` is an exponentially fast algorithm, $\alpha_{\text{tol}}$ can reach machine precision in a very few steps. For example in our implementation we set $\alpha_{\text{tol}} = 10^{-16}$ and this is achieved in 76 steps of `GoldenSearch`.

## C. Additional algorithmic details and sub-routines for `DopeWolfe`

In this section, we provide additional details for `DopeWolfe` (Algorithm 2). We begin by providing the complete pseudocode for the three undefined sub-routines of `DopeWolfe` (`UpdateInverse`, `GoldenSearch`, `UpdateLogDet`) in Algorithm 4. Next we provide explanation for various algorithmic choices made in `DopeWolfe`.

### C.1. Low-rank update for the $\log \det$ objective

As mentioned in Section 5.2 computing objective function values involves computing the $\log \det$ of a $d \times d$ matrix which has a worst case complexity of $O(d^3)$. However, we simplify this computation by noting that $V_\pi$ is only updated with the $r = \binom{K}{2}$ rank matrix $A_S A_S^\top$ (for some $S$). Since $A_S \in \mathbb{R}^{r \times d}$ this is a low-rank matrix when $r \ll d$, which allows us to simplify the relevant change in objective function value after the convex combination with a step size $\alpha$ as follows. For the sake of simplicity, we denote $\widetilde{V}^{\text{inv}} = (V_\pi)^{-1}/(1-\alpha)$ and $\widetilde{A} = \sqrt{\alpha}A_S$. Then,

$$g((1-\alpha) \cdot \pi + \alpha \cdot e_S) - g(\pi)$$
$$= \log \det\left((1-\alpha)V_\pi + \alpha A_S A_S^\top\right) - \log \det(V_\pi)$$
$$= \log \det\left((1-\alpha)(\mathbf{I}_{d \times d} + \widetilde{A}\widetilde{A}^\top \widetilde{V}^{\text{inv}})V_\pi\right) - \log \det(V_\pi)$$
$$= d\log(1-\alpha) + \log \det(\mathbf{I}_{d \times d} + \widetilde{A}\widetilde{A}^\top \widetilde{V}^{\text{inv}})$$
$$= d\log(1-\alpha) + \log \det(\mathbf{I}_{r \times r} + \widetilde{A}^\top \widetilde{V}^{\text{inv}} \widetilde{A}), \quad (22)$$

where the third equality follows from the facts that $\log \det(BC) = \log \det(B) + \log \det(C)$ and $\log \det(cB_{d \times d}) = d\log(c) + \log \det(B)$, and the third equality follows from the Weinstein–Aronszajn identity $\log \det(\mathbf{I} + BC) = \log \det(\mathbf{I} +$

---

**Algorithm 4** Sub-routines required for Algorithm 2

---

1   **Sub-routine** `UpdateInverse`$(V^{\mathrm{inv}}, A, \alpha)$**:**

2   $\quad$ Set $r = \binom{K}{2}$, $\widetilde{V}^{\mathrm{inv}} = V^{\mathrm{inv}}/(1-\alpha)$, and $\widetilde{A} = \sqrt{\alpha}A$

3   $\quad$ $V_+^{\mathrm{inv}} = \widetilde{V}^{\mathrm{inv}} - V^{\mathrm{inv}}\widetilde{A}\Big(\mathbf{I}_{r \times r} + \widetilde{A}^T V^{\mathrm{inv}} \widetilde{A}\Big)^{-1} \widetilde{A}^T V^{\mathrm{inv}}$

4   $\quad$ **return** $V_+^{\mathrm{inv}}$

5   **Sub-routine** `GoldenSearch`$(V^{\mathrm{inv}}, A, \alpha_{\mathrm{tol}})$**:**

6   $\quad$ Set $\varphi = (\sqrt{5}+1)/2, \alpha_a = 0, \alpha_h = \alpha_b = 1$

7   $\quad$ Initialize:

8   $\quad$ $\alpha_c = \alpha_a + \alpha_h \varphi^{-2}, \alpha_d = \alpha_a + \alpha_h \varphi^{-1}$

9   $\quad$ $V_c^{\mathrm{ld}} = \texttt{UpdateLogDet}(V^{\mathrm{inv}}, A, \alpha_c)$
$\quad$ $V_d^{\mathrm{ld}} = \texttt{UpdateLogDet}(V^{\mathrm{inv}}, A, \alpha_d)$

10  $\quad$ **while** $|\alpha_a - \alpha_b| \geq \alpha_{\mathrm{tol}}$ **do**

11  $\quad\quad$ $\alpha_h = \alpha_h \varphi^{-1}$

12  $\quad\quad$ **if** $V_c^{\mathrm{ld}} > V_d^{\mathrm{ld}}$ **then**

13  $\quad\quad\quad$ $\alpha_b, \alpha_d, V_d^{\mathrm{ld}}, \alpha_c = \alpha_d, \alpha_c, V_c^{\mathrm{ld}}, \alpha_a + \alpha_h \varphi^{-2}$

14  $\quad\quad\quad$ $V_c^{\mathrm{ld}} = \texttt{UpdateLogDet}(V^{\mathrm{inv}}, A, \alpha_c)$

15  $\quad\quad$ **else**

16  $\quad\quad\quad$ $\alpha_a, \alpha_c, V_c^{\mathrm{ld}}, \alpha_d = \alpha_c, \alpha_d, V_d^{\mathrm{ld}}, \alpha_a + \alpha_h \varphi^{-1}$

17  $\quad\quad\quad$ $V_d^{\mathrm{ld}} = \texttt{UpdateLogDet}(V^{\mathrm{inv}}, A, \alpha_d)$

18  $\quad$ **return** $(\alpha_a + \alpha_b)/2$

19  **Sub-routine** `UpdateLogDet`$(V^{\mathrm{inv}}, A, \alpha)$ **:**

20  $\quad$ Set $r = \binom{K}{2}$, $\widetilde{V}^{\mathrm{inv}} = V^{\mathrm{inv}}/(1-\alpha)$, and $\widetilde{A} = \sqrt{\alpha}A$

21  $\quad$ $V_+^{\mathrm{ld}} = d\log(1-\alpha) + \log\det\Big(\mathbf{I}_{r \times r} + \widetilde{A}^\top \widetilde{V}^{\mathrm{inv}} \widetilde{A}\Big)$

22  $\quad$ **return** $V_+^{\mathrm{ld}}$

---

$CB$) (Pozrikidis, 2014). `GoldenSearch` sub-routine (Line 5 of Algorithm 4) in turn uses the `UpdateLogDet` sub-routine (Line 19 of Algorithm 4) which implements (22) assuming the access to $V_\pi^{-1}$. Since (22) computes $\log\det$ of an $r \times r$ matrix we reduce complexity of computing change in objective function values from $O(d^3)$ to $O(r^3 + rd^2)$. Note that one can further reduce the total complexity of `GoldenSearch` from $O(\log_\varphi(\alpha_{\mathrm{tol}}^{-1})(r^3 + rd^2))$ to $O(\log_\varphi(\alpha_{\mathrm{tol}}^{-1}) + r^3 + rd^2)$ by computing eigenvalues $\{\lambda_i\}$ of $A_{S_t}^\top V_{\pi_t}^{-1} A_{S_t}$ once and then computing the change in objective function values as $d\log(1-\alpha) + \sum_i^r \log(1+\alpha\lambda_i)$, however this approach may be numerically less stable.

### C.2. Low-rank inverse update

Here we expand on how `DopeWolfe` computes the inverse of the $d \times d$ matrix $V_{\pi_t}$ used in `PartialGrad` (Line 12 of Algorithm 2) and `UpdateLogDet` (Line 19 of Algorithm 4) sub-routines. Naively implementing it at every iteration incurs a cost of $O(d^3)$ every iteration. Instead, `UpdateInverse` sub-routine (Line 1 of Algorithm 4) is used in `DopeWolfe` to iteratively update the matrix $V^{\mathrm{inv}}$ storing the inverse as follows

$$
\begin{aligned}
V_+^{\mathrm{inv}} &= ((1-\alpha)V_\pi + \alpha A_S A_S^\top)^{-1} \\
&= ((\widetilde{V}^{\mathrm{inv}})^{-1} + \widetilde{A}\widetilde{A}^\top)^{-1} \\
&= \widetilde{V}^{\mathrm{inv}} - \widetilde{V}^{\mathrm{inv}}\widetilde{A}(\mathbf{I}_{r \times r} + \widetilde{A}^\top \widetilde{V}^{\mathrm{inv}}\widetilde{A})^{-1}\widetilde{A}^\top \widetilde{V}^{\mathrm{inv}},
\end{aligned} \tag{23}
$$

where we used the notations from Section 5.2 and the second equality used the Woodburry matrix inversion identity (Guttman, 1946; Woodbury, 1950) with $r = \binom{K}{2}$. When $r \ll d$, this update rule improves the complexity of finding the inverse to $O(r^3 + rd^2)$. Similarly $V_0^{\mathrm{inv}}$ is also initialized using the same `UpdateInverse` sub-routine (Line 3 of

Algorithm 2) because (for $\gamma < 1$):

$$
\begin{aligned}
V_0^{\mathrm{inv}} &= (A_{S_{-1}} A_{S_{-1}}^\top + \gamma \mathbf{I}_{d \times d})^{-1} \\
&= (\gamma \mathbf{I}_{d \times d} + (1 - \gamma) A_{S_{-1}} A_{S_{-1}}^\top / (1 - \gamma))^{-1} \\
&= \mathtt{UpdateInverse}(\mathbf{I}_{d \times d}, A_{S_{-1}} / (1 - \gamma), 1 - \gamma)
\end{aligned}
\tag{24}
$$

Therefore even initialization only costs $O(r^3 + rd^2)$ compute.

### C.3. Sparse iterate update

Finally we discuss how we store and update the large $\binom{N}{K}$ dimensional iterate $\pi_t$ (Line 9 of Algorithm 2). If we initialize `DopeWolfe` with a sparse $\pi_0$ and if $T_{\mathrm{od}} \ll \binom{N}{K}$, then its iterates are sparse since the update rule $\pi_{t+1} = (1 - \alpha_t) \cdot \pi_t + \alpha_t \cdot e_{S_t}$ consists of the one-hot probability vector $e_{S_t}$. This implies that maintaining $\pi_t$ as a $\binom{N}{K}$ dimensional dense vector is unnecessary and expensive. Therefore, `DopeWolfe` maintains it as sparse vector $\pi_t^{(sp)}$ (`scipy.sparse.csc_array`[3]) of size $\binom{N}{K}$. This begs the question how `DopeWolfe` maps a subset $S \in \mathcal{S}$ to an index of $\pi_t^{(sp)}$. `DopeWolfe` answers this by using the combinadics number system of order $K$[4] which defines a bijective mapping, $\mathcal{C}_K : \mathcal{S} \to [\binom{N}{K}]$, from the collection of subsets of size $K$ to integers. Therefore our update rule in terms of $\pi_t^{(sp)}$ translates to $\pi_{t+1}^{(sp)} = (1 - \alpha_t) \cdot \pi_t^{(sp)} + \alpha_t \cdot e_{\mathcal{C}_K(S_t)}^{(sp)}$, where $e_{\mathcal{C}_K(S_t)}^{(sp)}$ is the basis vector for the $\mathcal{C}_K(S_t)$-th dimension. These modifications improves the worst case complexity of updating and maintaining $\pi_t$ after $t$ iterations from $O(\binom{N}{K})$ to only $O(t)$.

## D. Additional Experimental Details and Results

In this section we provide additional experimental details and results that were omitted from Section 6.

We experiment with (a) BEIR-COVID[5], (b) Trec Deep Learning (TREC-DL)[6], and (c) NECTAR[7] datasets. These are question-answering datasets, where the task is to rank answers or passages by relevance to the question. Each question in BEIR-COVID and TREC-DL has a hundred potential answers. Each question in NECTAR has seven potential answers. All the following experiments are conducted on 3.5 GHz 3rd generation Intel Xeon Scalable processors with 128 vCPUs and 1TB RAM. Code for our experiments is available at `https://github.com/tkkiran/DopeWolfe`.

### D.1. Synthetic Feedback Setup

In this section we provide additional experimental details and results that were omitted from Section 6.2.

We first compute 384-dimensional dense BERT embeddings (Reimers & Gurevych, 2019) for each question and answer, and then reduce the embedding dimensions to 10 by fitting UMAP (McInnes et al., 2018) on the answers in a dataset. The same UMAP transformation is applied to questions to get 10-dimensional embedding of the questions. Let $q$ and $a$ be the projected embeddings of a question and an answer to it. Then, we consider the outer product $vec(qa^T)$ as the feature vector of the question-answer pair. Its length is $d = 100$. We choose a random $\theta^* \in \mathbb{R}^{100}$ to generate feedback in (1).

The output of `DopeWolfe` is a distribution over all $K$-subsets of the original set of $N$ items. To show the efficacy of `DopeWolfe`, we compare it with the following baselines:

1. **Uniform:** This approach chooses $K$-subsets at random with equal probability.

2. **DBSCAN:** We apply DBSCAN clustering (Ester et al., 1996) over features of all $K$-subsets, each of which is a concatenation of the features of the items in it. DBSCAN has a distance hyper-parameter $\epsilon$ that has a major impact on the clustering. To select it, we evaluate several clustering $\epsilon \in \{10^{-5}, 10^{-4}, \ldots, 1\}$, and choose the one that results in the fewest clusters; and then define uniform distribution over the cluster centers. This can be viewed as the sparsest distribution of core samples or centroids.

---

[3]`https://scipy.org/`
[4]`https://en.wikipedia.org/wiki/Combinatorial_number_system`
[5]https://huggingface.co/datasets/BeIR/trec-covid
[6]https://microsoft.github.io/msmarco/TREC-Deep-Learning
[7]https://huggingface.co/datasets/berkeley-nest/Nectar

3. $m$**-medoids:** We set $m$ to the size of the support of the distribution from `DopeWolfe`. Then, we run the $m$-medoids algorithm [8] over the same features as in DBSCAN. Finally, we select $K$-subsets associated with (centroids) at random with equal probabilities. Note that this approach has extra information in terms of $m$ which other methods do not have.

We consider 1 random question in BEIR-COVID and TREC-DL datasets. The number of 2-way subsets and 3-way subsets for one question with hundred answers are $\binom{100}{2} = 4950$ and $\binom{100}{3} = 161700$, respectively. We consider random 30,000 questions in NECTAR dataset. The number of 2-way subsets and 3-way subsets, where each question has seven answers, are $30000\binom{7}{2} = 630000$ and $30000\binom{7}{3} = 1050000$, respectively. Given a sample size $T$, we fit a PL ranking model. $m$-medoids fails for the NECTAR dataset due to excessive memory requirements (2 TB and 8 TB of RAM, respectively).

In Figure 4, we show the mean NDCG metric (higher is better) for the experiment in Section 6.2 for $K = 2, 3$ on the three datasets. The observations are consistent with Section 6.2, where `DopeWolfe` achieves better ranking performance than baselines.

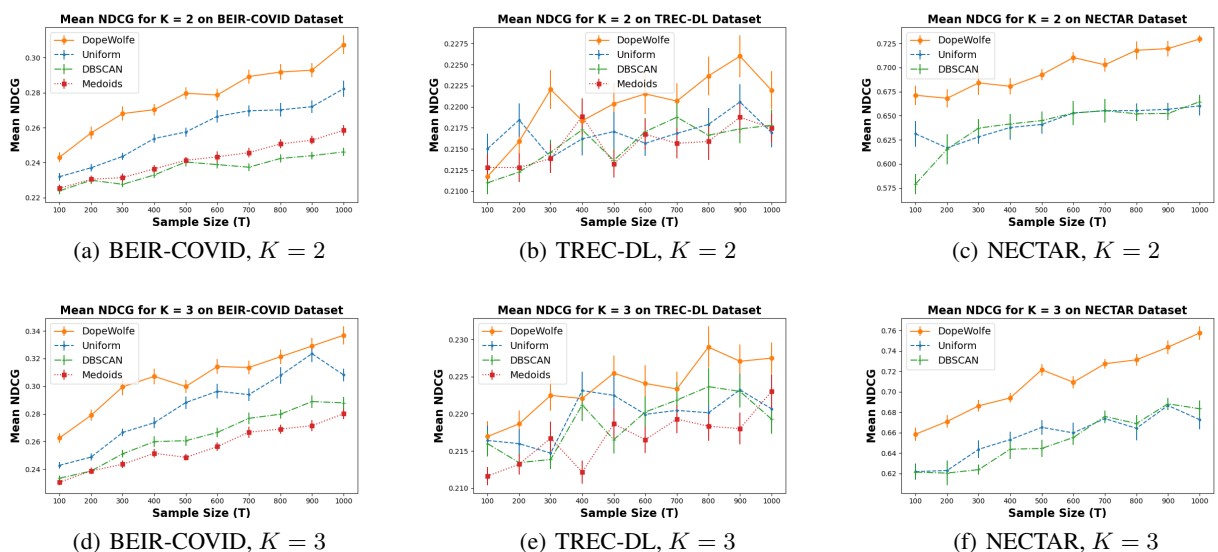

(a) BEIR-COVID, $K = 2$     (b) TREC-DL, $K = 2$     (c) NECTAR, $K = 2$

(d) BEIR-COVID, $K = 3$     (e) TREC-DL, $K = 3$     (f) NECTAR, $K = 3$

*Figure 4.* Mean NDCG metric (higher is better) on BEIR-COVID, TREC-DL and NECTAR datasets with synthetic feedback.

We next show the ranking loss (2) (lower is better) and NDCG metric (higher is better) for $K = 4$ for BEIR-COVID and TREC-DL datasets in Figure 5 for the synthetic setup (Section 6.2. Note that, for NECTAR dataset, since the number of answers for each question are only seven, the number of possible subsets are same for $K = 3$ and $K = 4$; thus, we ignore NECTAR dataset for $K = 4$. We again observe that `DopeWolfe` achieves better performance than baselines on the ranking task.

### D.2. Real Feedback Setup

In this section, we provide more details on the experimental choices for the experiment with real feedback (Section 6.3).

We use real feedback from TREC-DL and BEIR-COVID datasets. Each question in TREC-DL and BEIR-COVID datasets has 100 potential answers. The answers are ranked. When a learning algorithm queries a subset of $K$ answers, the feedback is their ranked order in the dataset. We vary $K$ at a fixed value of $T$ and observe how the metrics change.

Similarly to Section 6.2, we evaluate the feedback elicitation methods by assessing the downstream performance of the learned PL model. Specifically, we compare `DopeWolfe` to Uniform sampling for feedback elicitation. DBSCAN and $m$-medoids are excluded from this comparison due to their inability to scale effectively for this problem size.

For both BEIR-COVID and TREC-DL datasets, since the feedback is real, we use more representative 1024 dimensional BGE-M3 embedding (Chen et al., 2024) to ensure that the features are rich and reasonable enough to capture the true

---

[8]https://tinyurl.com/scikit-learn-m-medoids

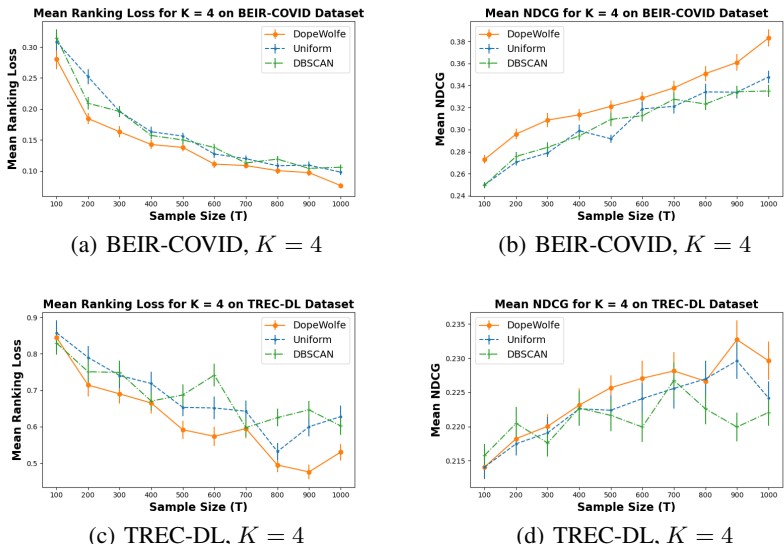

*Figure 5.* BEIR-COVID and TREC-DL datasets with synthetic feedback: Mean KDtau and NDCG metric on Datasets for $K = 4$. On average, `DopeWolfe` achieves better performance than Uniform sampling and DBSCAN.

*Table 1.* Run time comparison: `DopeWolfe` vs DBSCAN.

| Method | Time in Seconds |
|---|---|
| `DopeWolfe` at 1000 iterations | 1646.70 |
| DBSCAN ($\epsilon = 10^{-5}$) | 37.52 |
| DBSCAN ($\epsilon = 10^{-4}$) | 37.46 |
| DBSCAN ($\epsilon = 10^{-3}$) | 39.63 |
| DBSCAN ($\epsilon = 10^{-2}$) | 69.28 |
| DBSCAN ($\epsilon = 10^{-1}$) | 1346.28 |
| DBSCAN ($\epsilon = 1$) | 3397.21 |

ranking (realizable ranking problem). We then apply Principal Component Analysis (PCA) (Abdi & Williams, 2010) so that the feature matrix is full rank. We end up with 98 and 78 features for BEIR-COVID and TREC-DL datasets, respectively. We then run `DopeWolfe` on the datasets by setting $\gamma = 10^{-6}$, $\alpha_{\text{tol}} = 10^{-16}$, and $R = \min(\binom{100}{K}, 10^5)$ and initializing it with a uniformly chosen single subset, i.e. $\text{nnz}(\pi_0) = 1$. We find that the `DopeWolfe` converges in 200 iterations for BEIR-COVID and in 1500 iterations for TREC-DL. Since the feedback is real and does not change whenever we sample a new $K$-way subset for feedback, we pick the top $K$-way subsets that have the highest probability mass in the sampling distribution obtained through `DopeWolfe`. Note that due to significant amount of ranking ties in the BEIR-COVID dataset, we use the MLE for the pairwise rank breaking of the observations (Negahban et al., 2018, Equation 39, Page 25) instead of the $K$-wise MLE (3). We found that choosing only top-25 $K$-way subsets for BEIR-COVID works well, possibly due to the simpler nature of the problem as a consequence of the ties. For TREC-DL dataset, we choose top-500 $K$-way subsets. Lastly, we run the Plackett-Luce model's maximum likelihood estimation process using gradient descent with Barzilai-Borwein stepsize rule (Barzilai & Borwein, 1988) with min and max stepsize clip values of $10^{-8}$ and $5 \times 10^4/10^3$ for BEIR-COVID/TREC-DL, which converges in 200/1000 iterations for BEIR-COVID/TREC-DL. All the reported metrics provide mean and standard error over over 10 trials.

The ranking loss (2) results for this experiment is shown in Figure 3. We also show NDCG@10 for this experiment in Figure 6. We choose NDCG@10 to show the accuracy at the top of the ranking. In the NDCG computation, we use linear gain function for BEIR-COVID, and an exponential gain with temperature 0.1 for TREC-DL with the aim of differentiating its very close score values. We see that `DopeWolfe` performs better than the baselines for the ranking tasks in both datasets.

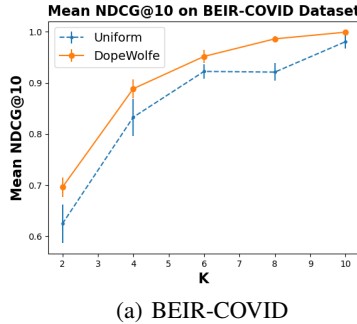
(a) BEIR-COVID

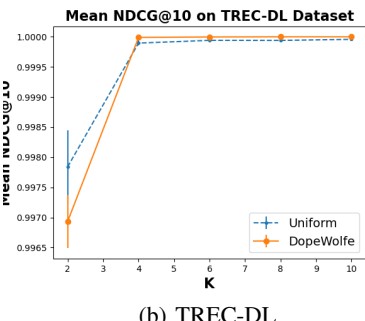
(b) TREC-DL

*Figure 6.* BEIR-COVID and TREC-DL datasets with real feedback: Models learned through ranking feedback collected on `DopeWolfe` samples achieve higher NDCG@10 than the ones learned on Uniformly selected samples.

### D.3. Practical Run Time Comparison

We now compare the run time of `DopeWolfe` with DBSCAN. Although the running times for `DopeWolfe` and DBSCAN are not directly comparable due to different deciding parameters—`DopeWolfe` relies on the number of Frank-Wolfe iterations, while DBSCAN depends on the distance threshold ($\epsilon$). We chose DBSCAN as a baseline clustering algorithm, because it does not require an input parameter specifying distribution sparsity (e.g., $m$-medoids requiring $m$), and it provides cluster centers as actual data points.

Nonetheless, in Table 1, we present run-time comparisons for `DopeWolfe` with 1000 Frank-Wolfe iterations and DBSCAN with distance thresholds varying between $[10^{-5}, \ldots, 1]$, as explained in Section 6.2 for the BEIR-COVID dataset with $K = 3$, corresponding to 161,700 possible subsets. We observe that DBSCAN takes twice the time of our proposed method for $\epsilon = 1$. The total run-time for the DBSCAN-based baseline is 4927.38 seconds; whereas ours is only 1646.70 seconds. Although we used 1000 iterations in all our experiments, convergence is typically achieved in around 100 iterations, yielding similar performance, which only takes 164.67 seconds. This demonstrates that our proposed method not only achieves superior performance but also is computationally efficient.

### D.4. Additional baseline

The closest related work from the bandit baselines in Appendix A is Saha & Gopalan (2019a) and we compare to it in the following experiments. Since both of their algorithms perform comparably, we implemented simpler Beat-the-Pivot (for TR feedback) as follows:

1. We give Beat-the-Pivot the optimal pivot (best item). This simplifies our implementation while giving Beat-the-Pivot a slight advantage.

2. We divide all items into groups of size $K - 1$ and add the pivot to each group.

3. We go over the groups in a round robin and get ranking feedback on them. For each item in each group, we estimate the fraction of times that it beats the pivot.

4. We sort the items by the fraction of times that they beat the pivot in a descending order. This yields the final ranked list.

We report our results in the synthetic feedback setting (Section 6.2) in Tables 2 and 3.

We observe that Beat-the-Pivot performs worse than any of our current baselines (compare to Figure 1 on page 7 and Figure 3 in Appendix). This is because the approach is not contextual. Specifically, since the relevance of each item is estimated independently, Beat-the-Pivot is less statistically efficient than the methods that utilize features. Due to this shortcoming, we also do not run experiments on the NECTAR dataset, which has 30k separate lists. Even with a total of 1000 samples, the number of samples allotted to a list could be 0 and non-contextual algorithms cannot share information across lists.

| Dataset | K | 100 | 200 | 300 | 400 | 500 | 600 | 700 | 800 | 900 | 1000 |
|---------|---|-----|-----|-----|-----|-----|-----|-----|-----|-----|------|
| BEIR-COVID | 2 | $0.87 \pm 0.01$ | $0.82 \pm 0.01$ | $0.80 \pm 0.01$ | $0.78 \pm 0.01$ | $0.78 \pm 0.01$ | $0.76 \pm 0.01$ | $0.74 \pm 0.01$ | $0.73 \pm 0.01$ | $0.72 \pm 0.01$ | $0.70 \pm 0.01$ |
| BEIR-COVID | 3 | $0.83 \pm 0.01$ | $0.78 \pm 0.01$ | $0.75 \pm 0.01$ | $0.73 \pm 0.01$ | $0.71 \pm 0.01$ | $0.69 \pm 0.01$ | $0.67 \pm 0.01$ | $0.67 \pm 0.01$ | $0.65 \pm 0.01$ | $0.63 \pm 0.01$ |
| TREC-DL | 2 | $0.99 \pm 0.01$ | $0.97 \pm 0.01$ | $0.96 \pm 0.01$ | $0.97 \pm 0.01$ | $0.97 \pm 0.01$ | $0.96 \pm 0.01$ | $0.96 \pm 0.01$ | $0.95 \pm 0.01$ | $0.96 \pm 0.01$ | $0.96 \pm 0.01$ |
| TREC-DL | 3 | $0.97 \pm 0.01$ | $0.98 \pm 0.01$ | $0.96 \pm 0.01$ | $0.95 \pm 0.01$ | $0.96 \pm 0.01$ | $0.94 \pm 0.01$ | $0.94 \pm 0.01$ | $0.94 \pm 0.01$ | $0.93 \pm 0.01$ | $0.93 \pm 0.01$ |

*Table 2.* Mean ranking loss of Beat-the-Pivot.

| Dataset | K | 100 | 200 | 300 | 400 | 500 | 600 | 700 | 800 | 900 | 1000 |
|---------|---|-----|-----|-----|-----|-----|-----|-----|-----|-----|------|
| BEIR-COVID | 2 | $0.21 \pm 0.00$ | $0.21 \pm 0.00$ | $0.21 \pm 0.00$ | $0.21 \pm 0.00$ | $0.21 \pm 0.00$ | $0.21 \pm 0.00$ | $0.21 \pm 0.00$ | $0.22 \pm 0.00$ | $0.21 \pm 0.00$ | $0.21 \pm 0.00$ |
| BEIR-COVID | 3 | $0.21 \pm 0.00$ | $0.21 \pm 0.00$ | $0.21 \pm 0.00$ | $0.22 \pm 0.00$ | $0.21 \pm 0.00$ | $0.22 \pm 0.00$ | $0.22 \pm 0.00$ | $0.22 \pm 0.00$ | $0.22 \pm 0.00$ | $0.22 \pm 0.00$ |
| TREC-DL | 2 | $0.21 \pm 0.00$ | $0.21 \pm 0.00$ | $0.21 \pm 0.00$ | $0.21 \pm 0.00$ | $0.21 \pm 0.00$ | $0.21 \pm 0.00$ | $0.21 \pm 0.00$ | $0.21 \pm 0.00$ | $0.21 \pm 0.00$ | $0.21 \pm 0.00$ |
| TREC-DL | 3 | $0.21 \pm 0.00$ | $0.21 \pm 0.00$ | $0.21 \pm 0.00$ | $0.21 \pm 0.00$ | $0.21 \pm 0.00$ | $0.21 \pm 0.00$ | $0.21 \pm 0.00$ | $0.21 \pm 0.00$ | $0.21 \pm 0.00$ | $0.21 \pm 0.00$ |

*Table 3.* NDCG of Beat-the-Pivot.

