# OpenReview forum: "Comparing Few to Rank Many: Active Human Preference Learning Using Randomized Frank-Wolfe Method"
_ICML.cc/2025/Conference — ICML 2025 poster_

### Official Review · Reviewer_Q6wK · 2025-03-07

**Overall Recommendation:** 4

**Summary:**

This paper considers an N-item-ranking problem with K-way comparisons, where K<<N. The goal is to determine the optimal K-subset selection strategy to minimize the worst-case ranking error. Previous approaches to this problem are computationally infeasible due to the exponentially large number to consider. To address this, the authors propose an improved Frank-Wolfe algorithm that incorporates randomization, memorization, and sparse updates, making the computation tractable.

**Claims And Evidence:**

The claims in the paper are thoroughly supported at both the theoretical and experimental levels.

**Essential References Not Discussed:**

No.

**Experimental Designs Or Analyses:**

The experiments include two small-scale text retrieval datasets and one LLM alignment dataset, along with multiple baselines, both synthetic and real feedback settings. The appendix also provides additional ablation experiments and runtime comparisons. I believe the experimental setup is thorough and reliable.

**Methods And Evaluation Criteria:**

Yes.

**Other Comments Or Suggestions:**

- page 15, line 3, missing citation.
- page 7, section 6.1, line 5, missing reference.

**Other Strengths And Weaknesses:**

**Strengths:**
- The paper is well-written, with a clear and rigorous logical structure. Despite the extensive mathematical theory, it remains easy to follow.
- The theoretical section is very thorough, with both theoretical and experimental evidence supporting the claims, making the results highly credible.
- The problem addressed by the paper is highly meaningful, and DopeWolfe makes the previously intractable algorithm with exponential computational complexity practically usable.

**Weaknesses:**
- If the paper included experiments on RLHF fine-tuning with LLMs, it would enhance the practical applicability and make the paper's findings more convincing.

**Questions For Authors:**

I have no questions.

**Relation To Broader Scientific Literature:**

The key contribution is related to the K-way feedback selection strategy in N-item ranking problems, which is highly relevant to RLHF and LLM alignment.

**Theoretical Claims:**

No mistakes were found in the proofs.

---

> ### Author Rebuttal · Authors · 2025-03-31
>
> We want to thank the reviewer for a positive review, and recognizing both benefits and shortcomings of our paper. Our rebuttal is below. We will incorporate all comments of the reviewer in the next version of our paper. If you have additional concerns, please reach out to us to discuss them.
>
> We agree with the reviewer that LLM experiments would increase the appeal of the paper. However, we could not conduct them due to limited computational resources. To further support our claims and partially address your comment, we contacted Mukherjee et al. (2024) and implemented DopeWolfe on the Anthropic RLHF dataset in their paper. The plot of the experiment is at [Anthropic plot](https://anonymous.4open.science/r/DopeWolfe-ICML-anon-1AD6/DopeWolfe_Anthropic_ICML_rebuttal.png). We observe that the ranking losses of DopeWolfe and Dope are comparable. This means that DopeWolfe beats all baselines in Mukherjee et al. (2024). However, in terms of the wall-clock time, DopeWolfe is about $4$x times faster than Dope ($6.046$ seconds on average versus $24.924$). See Section **Additional Baselines** of the rebuttal to **Reviewer Z4RR** for more details.

---

### Official Review · Reviewer_GkKM · 2025-03-07

**Overall Recommendation:** 3

**Summary:**

This paper investigates how to select the optimal data points to learn the ground-truth reward model in RLHF. Specifically, compared to previous work [1], this paper extends the result from learning the ranking of $K$ actions from $K$-way feedback to learning the ranking of $N\geq K$ actions from $K$-way feedback.

[1] Mukherjee, Subhojyoti, et al. "Optimal design for human preference elicitation." Advances in Neural Information Processing Systems 37 (2024): 90132-90159.

**Claims And Evidence:**

Yes. The proof sketch is clear and convincing to me.

**Essential References Not Discussed:**

I do not find any essential references not discussed.

**Experimental Designs Or Analyses:**

I've checked Section 6 and it sounds reasonable to me.

**Methods And Evaluation Criteria:**

The synthetic feedback and real feedback seem to be reasonable.

**Other Comments Or Suggestions:**

There is a missing reference on lines 358-359.

**Other Strengths And Weaknesses:**

Compared to [1], this paper does not include the scenario when multiple lists exist.

[1] Mukherjee, Subhojyoti, et al. "Optimal design for human preference elicitation." Advances in Neural Information Processing Systems 37 (2024): 90132-90159.

**Questions For Authors:**

Compared to [1], may you also compare the baselines including Dope, Avg Design, Clustered Design, and APO? Additionally, I would appreciate a comparison in the Anthropic dataset [3] used in [1].

I would raise the score if the authors could provide additional experiments.

[1] Mukherjee, Subhojyoti, et al. "Optimal design for human preference elicitation." Advances in Neural Information Processing Systems 37 (2024): 90132-90159.

[3] Bai, Yuntao, et al. "Training a helpful and harmless assistant with reinforcement learning from human feedback." arXiv preprint arXiv:2204.05862 (2022).

**Relation To Broader Scientific Literature:**

This paper could complement previous papers about how to compute the ground-truth reward from a dataset of rankings, such as [2]. Specifically, the previous paper considers how to compute the reward form a given dataset, while this paper focuses on how to collect the data to make the computation efficient.

[2] Zhu, Banghua, Michael Jordan, and Jiantao Jiao. "Principled reinforcement learning with human feedback from pairwise or k-wise comparisons." International Conference on Machine Learning. PMLR, 2023.

**Theoretical Claims:**

I checked the proof sketch in the main text but did not go through the proof in the appendix. The proof seemed to make sense according to the proof sketch in the main text.

---

> ### Author Rebuttal · Authors · 2025-03-31
>
> We want to thank the reviewer for acknowledging the contributions of our work and putting it in the context of prior works. Our rebuttal is below. We focus on major issues and will incorporate all comments of the reviewer in the next version of our paper. If you have additional concerns, please reach out to us to discuss them.
>
> ### **We Do Not Handle Multiple Lists**
>
> This is a writing issue. We sketch how to represent multiple lists in the last paragraph of Section 2. However, we failed to explicitly state that the algorithm remains unchanged when solving such problems. This is because it does not matter how the plausible subsets of items in DopeWolfe arise. They can be $K$-subsets of a single list or multiple.
>
> In fact, we already conducted an experiment like this in our paper. The Nectar dataset in Section 6.1 has $30000$ lists, each with $7$ choices. For $K$-way comparisons, we optimize over $30000 {7 \choose K}$ subsets of items. This is what $K = 2$ and $K = 3$ in Figure 1 are. Even for $K = 2$, this is $630000$ subsets and two orders of magnitude more than in Mukherjee et al. (2024). We describe this in lines 949-953 (Appendix D.1).
>
> ### **Additional Baselines**
>
> To alleviate your concerns about the lack of comparison to Dope and baselines in Mukherjee et al. (2024), we contacted Mukherjee et al. (2024) and obtained their code. The code differs from their paper as follows:
>
> * Dope is implemented using the Frank-Wolfe method where the linear maximization oracle (LMO) is a linear program.
> * The Nectar experiment is on $500$ lists. The code crashes for $2000$ lists used in Mukherjee et al. (2024) and they did not resolve this issue before we submitted this rebuttal.
>
> We implemented DopeWolfe in their code base as follows:
>
> * LMO is solved greedily, as in (8), on 10% of randomly sampled subsets of items.
> * Line search is implemented using golden-section search.
>
> The plots for the Anthropic and Nectar experiments are at [Anthropic plot](https://anonymous.4open.science/r/DopeWolfe-ICML-anon-1AD6/DopeWolfe_Anthropic_ICML_rebuttal.png) and [Nectar plot](https://anonymous.4open.science/r/DopeWolfe-ICML-anon-1AD6/DopeWolfe_Nectar_ICML_rebuttal.png), respectively. We observe that the ranking losses of DopeWolfe and Dope are comparable. This means that DopeWolfe beats all baselines in Mukherjee et al. (2024). However, there is a difference in the wall-clock time:
>
> * **Anthropic:** DopeWolfe is about $4$x times faster than Dope ($6.046$ seconds on average versus $24.924$).
> * **Nectar:** DopeWolfe is about $3$x times faster than Dope ($0.571$ seconds on average versus $1.826$).

---

> > ### Comment · Reviewer_GkKM · 2025-04-03
> >
> > Thank you for your response! I will raise the score to 3.

---

### Official Review · Reviewer_4hjz · 2025-03-14

**Overall Recommendation:** 3

**Summary:**

This paper focuses on the problem of RL using human ranking feedback. In particular, the goal is to learn the ranking of N items using K-way ranking feedback. To solve such a problem in an active learning fashion, the learner needs to first compute a design via Frank-Wolfe (FW), sample queries according to the design, estimate utility for each item, and estimate the ranking.

Previous work by Mukherjee et al. (2024) solved the problem where N=K. This paper extends Mukherjee et al. (2024) to the case when N>>K. The key challenge in this extension is that the FW gradient has a dimension of (choose N K), which is computationally challenging for modern computers.

There are 2 contributions: (1) analyzing the ranking loss by extending Mukherjee et al. (2024) to show that the original method from Mukherjee et al. (2024) is still valid for the case when N>>K if ignoring the computation challenge; (2) proposes a computationally efficient method, called "DopeWolfe" by integrating randomized FW (Kerdreux et al., 2018) and golden-section search method (Kiefer, 1953) to approximately solve FW and reduce the computation with a factor of (choose N K) to (choose N 2). Also, the authors prove that after O(max(1, (choose N K)/R)/ε) iterations, the error is upper bounded by ε.

The authors tested the proposed method vs 3 baseline methods on 3 real-world datasets. In the first experiment, the human preference vector, θ, is randomly sampled, and human feedback is simulated accordingly. In the second experiment, there is no ground truth θ (which makes the proposed method using the PL model misspecified), and the human feedback is taken from the dataset. The result shows that the proposed method with the MLE estimator achieves the lowest ranking loss.

## update after rebuttal
I appreciate the authors' Rebuttal in addressing my concerns. I have adjusted my score accordingly.

**Claims And Evidence:**

* Claim 1: Mukherjee et al. (2024) is valid for the case of N>>K when ignoring the computation challenge.
    * This is theoretically justified by bounding the ranking loss in Sec.3.3 and empirically justified in experiments to show that the pipeline achieved higher performance than others.
    * I have one concern about the empirical result. In the result in Sec.6.1 (using synthetic feedback), the proposed method significantly outperformed baselines (Fig.1). However, in the result in Sec.6.2 (using real feedback), the advantage of the proposed method, in comparison to the baseline is less than 0.1 for mean ranking loss (Fig.2). It might be helpful to justify this by explaining that this difference can be significant in real-world systems.

* Claim 2: when extending Mukherjee et al. (2024) to the case of N>>K, the FW is computationally challenging due to the computation complexity factor (choose N K). The proposed method resolves this by getting rid of the (choose N K).
    * This is theoretically justified by analyzing the computation complexity in Sec.5.1 and 5.2, by showing e.g., (choose N K) is reduced to (choose N 2). This is also empirically justified in experiments to show that the proposed method is able to solve the FW while vanilla FW won't work.
    * I have one concern about the theoretical result. Cor.4 shows that to solve the proposed algorithm to reach small error ε, it would need O(max(1, (choose N K)/R)/ε) iterations. So essentially, in each iteration, the proposed method is able to get rid of the (choose N K), but the overall optimization still needs (choose N K) iterations. I understand that there is no free lunch, and the empirical result seems to be working well, but it might be helpful for the authors to clarify this.

**Essential References Not Discussed:**

No.

**Experimental Designs Or Analyses:**

The experiment design and analysis make sense.

**Methods And Evaluation Criteria:**

Yes. The empirical study is based on 3 real-world datasets about text retrieval and LLM alignment. The synthetic-feedback experiment randomly selects θ, which makes sense because the problem is too complicated to identify the θ from the dataset. The real-feedback experiment directly uses human feedback and benchmarking the ranking loss, which makes sense.

**Other Comments Or Suggestions:**

Typo in Sec.6.1: `form sampling (see ??)`

**Other Strengths And Weaknesses:**

This paper is well written, with good theoretical and empirical justification. The technical contribution is not so "surprising" as it integrates several existing FW variations to solve an optimal design problem computationally efficiently. But the extension is well-motivated and the proposed method performed empirically better than baselines. So I think it is a good contribution.

**Questions For Authors:**

In Prop.2. I am curious why there is the extra additive d in the exponent. I checked that it also appears in Thm.6 in Mukherjee et al. (2024) (https://arxiv.org/pdf/2404.13895). Mukherjee et al. (2024) said that `This dependence is similar to Theorem 2 of Azizi et al. [6] for fixed-budget best-arm identification in

**Relation To Broader Scientific Literature:**

The result can be useful for learning human preferences from ranking feedback for modern AI systems, such as LLMs, if the set of prompts is fixed and known, as demonstrated by the empirical result.

**Theoretical Claims:**

I didn't check the proofs. Prop.1 adapts from Thm.5 of Mukherjee et al. (2024), and Prop.2 adapts from Thm.6 of Mukherjee et al. (2024).

One concern is that Assumption 1 assumes that ||θhat||_2<=1. I am wondering how this can be ensured during the MLE?

---

> ### Author Rebuttal · Authors · 2025-03-31
>
> We want to thank the reviewer for a positive review that discusses both the new aspects of our work and potential shortcomings. Our rebuttal is below. We focus on major issues and will incorporate all comments of the reviewer in the next version of our paper. If you have additional concerns, please reach out to us to discuss them.
>
> ### **Empirical Concern in Claim 1**
>
> We acknowledge that the setting of real feedback is potentially more challenging because the optimized model is misspecified (unrealizable). However, we do not think that Figure 2 shows significantly worse results than Figure 1. First, note that the figures are plotted differently: all plots in Figure 1 are functions of a sample size for fixed $K$, while Figure 2 is a function of $K$ for a fixed sample size ($500$ in TREC-DL, for instance). Therefore, they are not comparable. Second, the corresponding experiments use different embeddings. Given all the differences, the relative difference in ranking losses of Uniform and DopeWolfe in the same plot is one natural metric. This is consistently 20% in Figure 2 and similar to many relative differences in Figure 1.
>
> ### **Theory Concern in Claim 2**
>
> Thank you for understanding the computational feasibility perspective that we tried to convey in the last two paragraphs of Section 5. We reiterate that DopeWolfe was designed to address the concern that the naive implementation of Dope is infeasible even for small problems (Section 4).
>
> ### **Ensuring Assumption 1 During MLE**
>
> We can ensure $\|\|\hat{\theta}\|\|_2 \leq 1$ during MLE as follows. First, note that $\Theta = \\{\theta \in \mathbb{R}^d: \|\|\theta\|\|_2 \leq 1\\}$ is a convex set. Second, the MLE in (3) is a convex optimization problem. Therefore, we can minimize (3) on $\Theta$ using gradient descent with a projection step to $\Theta$.
>
> ### **Additive $d$ in Proposition 2**
>
> The additive $d$ in Mukherjee et al. (2024) arises at the end of the proof of Theorem 6 (Appendix A.6), because a high-probability upper bound on the model parameter error is $O(d + \log(1 / \delta))$. See their Lemma 9 (Appendix B) for the statement and proof of the bound.

---

### Official Review · Reviewer_Z4RR · 2025-03-15

**Overall Recommendation:** 2

**Summary:**

This paper proposes an algorithm to rank N items from K-way comparisons (K<<N), formulated through a D-optimal design objective. The authors develop “DopeWolfe,” a randomized Frank-Wolfe variant with sparse and low-rank updates to avoid high complexity of naive approaches. They prove a sublinear convergence rate for this randomized method under a logarithmically-homogeneous self-concordant barrier analysis. Empirical results on both synthetic and real IR datasets suggest that the proposed design framework improves ranking accuracy compared to uniform sampling or clustering-based methods.

**Claims And Evidence:**

Claims are supported by theoretical analysis.

**Essential References Not Discussed:**

None.

**Experimental Designs Or Analyses:**

**Data/Setup Concerns.** The paper’s experiments primarily center on two information-retrieval datasets, each with N=100≫K=2. While these retrieval tasks do involve preference judgments, they may not accurately reflect the complexities of human-generated comparisons found in large-scale response-based RLHF settings, where user preferences can be more nuanced or subjective. Moreover, for the LLM-alignment dataset, the paper mentions N=7 items across multiple lists—a rather small scale. In such a scenario, it seems more prior baselines (e.g., DOPE) could be applied and compared, since the problem size is not computationally prohibitive.

**Baselines.** On the “real data” setting, the paper omits the two baselines in synthetic setting, even though K remains small. Also, including baselines that is “smarter” than uniform sampling yet still computationally manageable would be more convincing. For instance, iterating through all data in random order and pairing.

**Hyperparameter Analysis.** More tuning or ablation studies on key hyperparameters will be helpful.

**Methods And Evaluation Criteria:**

A key assumption in the paper is that a global ranking over N items can be inferred solely by issuing multiple K-ranking queries—i.e., repeatedly asking annotators to compare small subsets of items. However, this setup becomes questionable when N is large and the total number of queries T times K exceeds N. In real-world systems, it may be more natural and cost-effective to collect absolute (or per-item) feedback for high-dimensional or large-scale ranking tasks, rather than repeated sampling of the same few items in multiple queries. The results in the experiment gets small ranking error when T times K exceeds.

**Other Comments Or Suggestions:**

Line 358 right column has a reference error (??)

**Other Strengths And Weaknesses:**

The problem setting is interesting but may not be very practical. The proposed algorithm nicely solve the problem and theoretical analysis is provided. But more real-world experiments are needed to support practical use.

**Questions For Authors:**

In practice, K-way queries might arrive in a sequential setting with noisy or incomplete feedback. How could DopeWolfe incorporate incremental updates as partial responses come in?

**Relation To Broader Scientific Literature:**

The paper situates itself in the broader context of RLHF and preference-based ranking. Its approach to active preference learning for
K-wise feedback is connected to D-optimal design, a classic concept in statistics but still underexplored in RLHF or large-scale ranking tasks. It generalizes prior active ranking work (some of which focuses on pairwise or small-scale scenarios) to the more general Plackett-Luce model for K-way comparisons. The use of a randomized Frank-Wolfe variant for log-determinant objectives extends earlier results on Lipschitz smooth objectives, bridging a gap in the continuous optimization literature.

**Theoretical Claims:**

The theorems appear consistent with known self-concordance analyses.

---

> ### Author Rebuttal · Authors · 2025-03-31
>
> We want to thank the reviewer for detailed feedback, and recognizing that we solved the problem that we set out to solve well. Our rebuttal is below. We focus on major issues and will incorporate all comments of the reviewer in the next version of our paper. If you have additional concerns, please reach out to us to discuss them.
>
> ### **Practical Motivation**
>
> $K$-wise comparisons naturally arise in many large-scale systems. For instance, online marketplaces frequently display a handful of items selected from a catalogue of billions of items to a user, and the user’s choice among those items can be viewed as a noisy ranking observation. These real-world interactions are well-modeled by the Plackett–Luce model used in our work (Negahban et al., 2018; [Buchholz et al., 2022](https://arxiv.org/pdf/2205.06024)).
>
> Recent developments in reinforcement learning with human feedback (RLHF) also suggest that relative feedback can yield better alignment than absolute feedback ([Christiano et al., 2017](https://arxiv.org/abs/1706.03741); [Bai et al., 2022](https://arxiv.org/abs/2204.05862)). It has also been shown that comparisons on longer lists ($K > 2$) can improve downstream reward models (Zhu et al., 2023b; [Liu et al., 2024](https://arxiv.org/abs/2402.01878)). This is one motivation for collecting the Nectar dataset used in Mukherjee et al. (2024).
>
> ### **Absolute Versus Relative Feedback**
>
> We acknowledge the concern that repeated $K$-wise comparisons may become costly. However, it is known that absolute feedback has its shortcomings as well ([Shah et al., 2014](https://arxiv.org/abs/1406.6618)). First, annotators often exhibit variable calibration when assigning absolute scores, making cross-annotator aggregation difficult. Second, comparison of items is often more natural and faster for humans. Finally, often we only have access to proxy signals (ad clicks or video watch time), which have an approximately monotonic relation to the real preference. A relative comparison requires only ordering rather than judging the precise value, mitigating calibration and proxy issues. Ultimately, whether the absolute or relative feedback is used is a design choice, and we believe that both should be studied.
>
> ### **Data and Setup Concerns**
>
> We would like to clarify that our Nectar experiment is not small scale. The dataset has $30000$ lists, each with $7$ choices. For $K$-way comparisons, we optimize over $30000 {7 \choose K}$ subsets of items. Even for $K = 2$, this is $630000$ subsets and two orders of magnitude more than in Mukherjee et al. (2024). We describe this in lines 949-953 (Appendix D.1).
>
> ### **Additional Baselines**
>
> To alleviate your concerns about the lack of comparison to Dope and baselines in Mukherjee et al. (2024), we contacted Mukherjee et al. (2024) and obtained their code. The code differs from their paper as follows:
>
> * Dope is implemented using the Frank-Wolfe method where the linear maximization oracle (LMO) is a linear program.
> * The Nectar experiment is on $500$ lists. The code crashes for $2000$ lists used in Mukherjee et al. (2024) and they did not resolve this issue before we submitted this rebuttal.
>
> We implemented DopeWolfe in their code base as follows:
>
> * LMO is solved greedily, as in (8), on 10% of randomly sampled subsets of items.
> * Line search is implemented using golden-section search.
>
> The plots for the Anthropic and Nectar experiments are at [Anthropic plot](https://anonymous.4open.science/r/DopeWolfe-ICML-anon-1AD6/DopeWolfe_Anthropic_ICML_rebuttal.png) and [Nectar plot](https://anonymous.4open.science/r/DopeWolfe-ICML-anon-1AD6/DopeWolfe_Nectar_ICML_rebuttal.png), respectively. We observe that the ranking losses of DopeWolfe and Dope are comparable. This means that DopeWolfe beats all baselines in Mukherjee et al. (2024). However, there is a difference in the wall-clock time:
>
> * **Anthropic:** DopeWolfe is about $4$x times faster than Dope ($6.046$ seconds on average versus $24.924$).
> * **Nectar:** DopeWolfe is about $3$x times faster than Dope ($0.571$ seconds on average versus $1.826$).
>
> Note that both Anthropic and Nectar are well-established RLHF datasets.

---

### Decision · Program_Chairs · 2025-05-01

**Decision:**

Accept (poster)

**Comment:**

In the context of active object ranking, this study addresses the problem of ranking $ N $ objects based on $ K $-way feedback. In each interaction, the learner presents a subset of $ K $ objects from the $ N $ total objects, which the user then orders. After $ T $ interactions, a ranking of the $ N $ items is generated, and its effectiveness is evaluated using ranking loss. The main contributions of this paper are: (i) an elegant formulation based on D-optimal design, and (ii) an efficient solution with convergence guarantees, supported by empirical validation through a variant of the randomized Frank-Wolfe algorithm.

Reviewers have noted that the paper is well-written (although there are a few typos). The primary concern raised by the reviewers is the absence of empirical comparisons with additional benchmarks and baselines. However, the authors effectively addressed these concerns during the rebuttal phase by conducting additional experiments.

In a revised version, I suggest including the authors' comments addressing the reviewers' concerns. Specifically, the experiments conducted on additional benchmarks and baselines (especially Dope) are informative. It would also be beneficial to discuss the overall complexity of the algorithm: while the per-iteration complexity is indeed low, the number of iterations required to ensure convergence is exponential in $ K $.